# Capturing the Temporal Dependence of Training Data Influence

**Jiachen T. Wang**[*]
Princeton University

**Dawn Song**
UC Berkeley

**James Zou**
Stanford University

**Prateek Mittal**
Princeton University

**Ruoxi Jia**[*]
Virginia Tech

## Abstract

Traditional data influence estimation methods, like influence function, assume that learning algorithms are permutation-invariant with respect to training data. However, modern training paradigms, especially for foundation models using stochastic algorithms and multi-stage curricula, are sensitive to data ordering, thus violating this assumption. This mismatch renders influence functions inadequate for answering a critical question in machine learning: How can we capture the dependence of data influence on the optimization trajectory during training? To address this gap, we formalize the concept of trajectory-specific leave-one-out (LOO) influence, which quantifies the impact of removing a data point from a specific iteration during training, accounting for the exact sequence of data encountered and the model's optimization trajectory. However, exactly evaluating the trajectory-specific LOO presents a significant computational challenge. To address this, we propose data value embedding, a novel technique enabling efficient approximation of trajectory-specific LOO. Specifically, we compute a training data embedding that encapsulates the cumulative interactions between data and the evolving model parameters. The LOO can then be efficiently approximated through a simple dot-product between the data value embedding and the gradient of the given test data. As data value embedding captures training data ordering, it offers valuable insights into model training dynamics. In particular, we uncover distinct phases of data influence, revealing that data points in the early and late stages of training exert a greater impact on the final model. These insights translate into actionable strategies for managing the computational overhead of data selection by strategically timing the selection process, potentially opening new avenues in data curation research.

## 1 Introduction

**Data influence estimation** aims to provide insights into the impact of specific data points on the model's predictive behaviors. Such understanding is crucial not only for model transparency and accountability (Koh & Liang, 2017) but also plays a significant role in addressing AI copyright debates (Deng & Ma, 2023; Wang et al., 2024) and facilitating fair compensation in data marketplaces (Tian et al., 2022). The majority of data influence estimation techniques focus on measuring the counterfactual impact of a training data point: *how would the model's behavior change if we removed a specific training data point?*

**LOO Influence.** This counterfactual impact is often characterized by the *Leave-One-Out* (LOO) influence, which has a long history and is frequently utilized in various fields such as robust statistics (Cook & Weisberg, 1980), generalization analysis (Bousquet & Elisseeff, 2002), and differential privacy (Dwork et al., 2006). Inheriting from this rich classical literature across various domains, the LOO influence in data influence studies is typically defined as $\mathrm{LOO}(z^*; z^{(val)}) := \ell(\mathcal{A}(\mathcal{D}), z^{(val)}) -$

---

[*]Correspondence to **Jiachen T. Wang** and **Ruoxi Jia** (`tianhaowang@princeton.edu`, `ruoxijia@vt.edu`).

$\ell(\mathcal{A}(\mathcal{D} \setminus \{z^*\}), z^{(val)})$, i.e., the model's loss change on a validation data $z^{(val)}$ when the training data point $z^*$ is removed from the training set $\mathcal{D}$. Here, $\mathcal{A}$ is the learning algorithm. For ease of analysis, traditional literature usually assumes that the learning algorithm $\mathcal{A}$ is permutation-invariant with respect to the training set $\mathcal{D}$, meaning that *the order of data points does not affect the learning outcome* (Bousquet & Elisseeff, 2002). This assumption holds for models with strongly convex loss functions trained to converge. Within this framework, researchers have developed efficient methods to approximate LOO. Influence function (Koh & Liang, 2017), which uses first-order Taylor expansion to estimate the LOO, emerging as the most prominent approach. Numerous follow-up works have further improved its scalability for large models and datasets (Guo et al., 2021; Schioppa et al., 2022; Grosse et al., 2023; Choe et al., 2024).

However, modern training algorithms, particularly those used for foundation models, increasingly deviate from the permutation-invariant assumption. This deviation arises from both the non-convex nature of neural networks and the multi-stage training curricula that do not run to convergence. In particular, due to the immense size of datasets, large language models (LLMs) often undergo just one training epoch, meaning each data point is encountered only once during training. Consequently, training data order significantly shapes the influence of data points on the final model (Epifano et al., 2023; Nguyen et al., 2024). Due to their underlying assumption of permutation-invariance, the order-dependence of data influence in modern training paradigms is not accurately reflected by influence functions. For example, they assign identical influence scores to duplicate training points, regardless of their position in the training sequence.

Therefore, in this work, we argue that designing a data influence estimation technique relevant to the modern ML context requires rethinking how the counterfactual impact should be defined. Towards that end, we formalize the concept of *trajectory-specific LOO*, which characterizes the loss change resulting from removing a data point from the specific iteration it is used during training. In contrast to the traditional LOO, trajectory-specific LOO explicitly accounts for the exact sequence of data encountered, considering the timing of a target training point being trained on. An accurate evaluation of trajectory-dependent LOO would enable us to answer many important questions that are impossible to address with influence functions. For instance, how does a data point's impact vary depending on its entry timing in the training process? How do later points affect the influence of earlier points?

However, exactly evaluating the trajectory-specific LOO presents a significant computational challenge. To address this, we introduce **data value embedding**, a novel data influence estimation framework designed for approximating trajectory-specific LOO. Our approach achieves several nice properties at the same time: **(1) accounting for training dynamics** and reflecting how the data order impacts model training; **(2) scale efficiently** to the setting of foundation models, and is faster than the current most efficient implementation of influence function; **(3) enable real-time attribution** for any query without necessitating model retraining or prior access to validation data.

**Technical novelty.** Our proposed *data value embedding* framework computes a compact representation for each data point that encapsulates the cumulative effect of subsequent training. The influence scores for any test instance can be approximated with a simple dot product operation between the test gradient and the data value embedding, enabling real-time computation of data influence scores. To improve the scalability of computing data influence embedding, we develop a suite of techniques for efficient computation and storage of data value embeddings. In particular, we introduce the *influence checkpointing* technique, which enables the parallel computation of data value embeddings at multiple checkpoints. This not only enhances computational efficiency but also allows tracking of how a fixed data point's value changes during the training process.

**Empirical insights.** Through data value embedding, we obtain several novel empirical insights into the training dynamics of foundation models. We identified three distinct regimes of data influence (Figure 1 (a)): a very brief high-influence region at the start, a much longer low-influence basin, and a region in the later training stage with gradually increasing influence, resuming to a high level. We show that performing online data selection solely in the early and late high-influence regions (less than half of the training duration) can achieve performance improvements on par with selecting data throughout the entire process (Figure 1 (b)). Moreover, performing data selection (Fan et al., 2024) only in the first very brief high-influence region, lasting less than 4% of the training duration, can achieve $\approx 50\%$ of the performance gain enabled by continuous selection. Since online data selection usually incurs significant computational costs, our findings suggest a viable way of managing this overhead by strategically timing the selection process. By focusing data

selection efforts on these critical phases, we can substantially improve training efficiency without compromising model performance. These temporal insights can potentially embark on new avenues of research on budget-limited data curation.

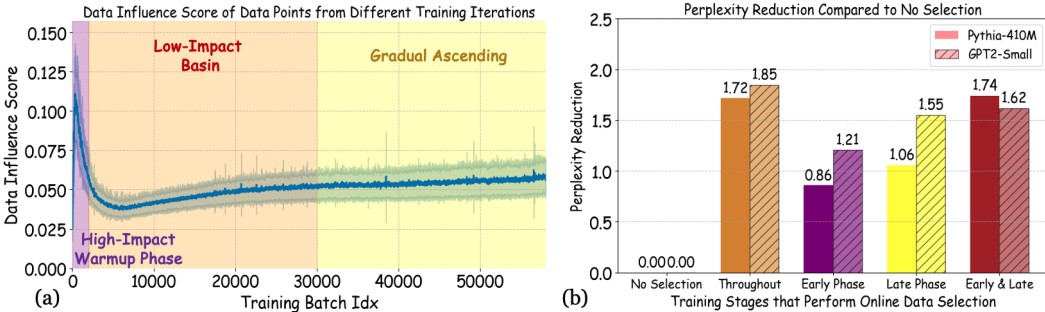

Figure 1: **(a)** Average data influence scores computed from data value embedding per training batch, measured against the final model's loss on Pile's validation set. Setting: Pythia-410M trained on 1% of Pile. **(b)** Comparison of online data selection strategies for training Pythia-410M on Pile. All strategies use gradient cosine similarity to Pile's validation set to select high-quality training batches (Fan et al., 2024), and only differ in the training stages during which advanced batch selection is applied (random selection otherwise).

## 2 TRAJECTORY-SPECIFIC LEAVE-ONE-OUT INFLUENCE

In this section, we formalize the definition of *trajectory-specific LOO* which was originally introduced in Hara et al. (2019) as 'SGD-influence'. Consider a data point $z^*$ that is included in the training process during the $t_s$-th iteration. Let $\mathcal{B}_t$ denote the training batch at iteration $t$. In standard SGD, the model parameters are updated as $\theta_{t+1} = \theta_t - \eta_t \sum_{z \in \mathcal{B}_t} \nabla \ell(\theta_t, z)$ for $t = 0, \ldots, T-1$, where $\eta_t$ is the learning rate at iteration $t$. We are interested in the change in the validation loss $\ell(\theta_T, z^{(val)})$ when the data point $z^* \in \mathcal{B}_{t_s}$ is removed from iteration $t_s$. In this counterfactual scenario, the parameter updates proceed as $\theta'_{t_s+1} = \theta_{t_s} - \eta_{t_s} \sum_{z \in \mathcal{B}_{t_s} \setminus \{z^*\}} \nabla \ell(\theta_{t_s}, z)$ and $\theta'_{t+1} = \theta'_t - \eta_t \sum_{z \in \mathcal{B}_t} \nabla \ell(\theta'_t, z)$ for $t = t_s + 1, \ldots, T-1$.

**Definition 1** (**Trajectory-Specific LOO** (Hara et al., 2019)). *The* trajectory-specific leave-one-out *for data point $z^*$ at iteration $t_s$ with respect to validation point $z^{(val)}$ is defined as*

$$TSLOO^{(t_s)}(z^*; z^{(val)}) := \ell(\theta'_T, z^{(val)}) - \ell(\theta_T, z^{(val)})$$

**Discussion.** `TSLOO` quantifies the change in validation loss resulting from removing $z^*$ during the specific training run determined by the sequence of mini-batches and random initialization. `TSLOO` explicitly depends on the timing of when the data is used and models the interaction effects between data points. For instance, it can show how the introduction of a certain type of example (e.g., a challenging edge case) might amplify or diminish the influence of previously seen, related examples. Moreover, identical data points contributing at different stages of training can receive different value scores. A data point introduced early in training might have a significantly different impact compared to the same point introduced later, as the model state evolves. However, traditional methods like influence functions do not capture these temporal dynamics. The influence function is defined as $\mathbf{IF}(z^*; z^{(val)}) := \nabla_\theta \ell(\theta, z^{(val)})^\top \mathbf{H}_\theta^{-1} \nabla_\theta \ell(\theta, z^*)$ where $\mathbf{H}_\theta$ is the Hessian with respect to the full training loss. Because $\mathbf{IF}$ depends solely on the final state of the model, it invariably assigns the same influence value to identical $z^*$s, regardless of their position in the training sequence.

**Related works (extended version in Appendix A).** Data attribution methods primarily fall into two categories: LOO-based methods and Shapley value-based methods. While Shapley value-based methods (Ghorbani & Zou, 2019) offer elegant theoretical interpretation, they typically require expensive model retraining, which limits their practical applicability. As a result, LOO-based methods such as influence functions (Koh & Liang, 2017) have gained more attention due to their computational efficiency. However, many studies have demonstrated that influence functions can be highly unreliable when applied to deep learning models (Basu et al., 2020; Bae et al., 2022; Epifano

et al., 2023). In this work, we argue that `TSLOO` provides a more appropriate attribution framework for deep learning, particularly in the context of foundation models. Various research communities have independently explored Taylor expansion-based technique (Section 3.1) for approximating `TSLOO` for different purposes (Hara et al., 2019; Zou et al., 2021; Evron et al., 2022; Wu et al., 2022; Luo et al., 2023; Wu et al., 2024; Ding et al., 2024). However, practical adoption has been hindered by computational demands. This work proposes a new method that overcomes the computational bottlenecks in approximating `TSLOO` for large-scale models.

## 3   DATA VALUE EMBEDDING

While trajectory-specific LOO offers clear benefits for understanding data influence in modern ML, its computation presents significant challenges. Exact computation is not feasible, as it would require removing a data point from a specific training iteration and re-initiating the entire training process. To address this challenge, we introduce the concept of *data value embedding*.

### 3.1   PRELIMINARY: UNROLLING THE EFFECT OF A TRAINING DATA POINT IN SGD

Recall that we denote the final model as $\theta_T$ and the counterfactual model as $\theta_T'$, which is obtained by removing $z^*$ from $t_s$-th training iteration. We introduce an interpolation between $\theta_T$ and $\theta_T'$ by defining $\theta_{t_s+1}(\varepsilon) := \theta_{t_s} - \eta_{t_s} \sum_{z \in \mathcal{B}_{t_s} \setminus \{z^*\}} \nabla \ell(\theta_{t_s}, z) - \eta_{t_s}(1 - \varepsilon) \nabla \ell(\theta_{t_s}, z^*)$ and $\theta_{k+1}(\varepsilon) = \theta_k(\varepsilon) - \eta_k \sum_{z \in \mathcal{B}_k} \nabla \ell(\theta_k(\varepsilon), z)$ for subsequent iterations. Note that $\theta_T(0) = \theta_T$ and $\theta_T(1) = \theta_T'$. Analogous to influence function-based approaches, we approximate the change in validation loss using a first-order Taylor expansion around $\varepsilon = 0$: $\ell(\theta_T', z^{(val)}) - \ell(\theta_T, z^{(val)}) \approx \nabla \ell(\theta_T, z^{(val)})^\top \frac{\partial \theta_T(\varepsilon)}{\partial \varepsilon}\Big|_{\varepsilon=0}$. Interestingly, the derivative $\frac{\partial \theta_T(\varepsilon)}{\partial \varepsilon}\Big|_{\varepsilon=0}$ satisfies a recursive relation detailed in Appendix C.1, and we can obtain a well-established approximation from the literature:

$$\ell(\theta_T', z^{(val)}) - \ell(\theta_T, z^{(val)}) \approx \eta_{t_s} \nabla \ell(\theta_T, z^{(val)})^\top \left[ \prod_{k=t_s+1}^{T-1} (\boldsymbol{I} - \eta_k \mathbf{H}_k) \right] \nabla \ell(\theta_{t_s}, z^*). \quad (1)$$

where $\mathbf{H}_k = \sum_{z \in \mathcal{B}_k} \nabla^2 \ell(\theta_k, z)$ is the Hessian and $\boldsymbol{I}$ is the identity matrix. In data attribution literature, this approximation in (1) first appears in Hara et al. (2019) and has also been utilized in Chen et al. (2021) and Bae et al. (2024). It estimates the influence of removing $z^*$ from the $t_s$-th iteration on the validation loss $\ell(\theta_T, z^{(val)})$ at the final iteration. The product term $\prod_{k=t_s+1}^{T-1} (\boldsymbol{I} - \eta_k \mathbf{H}_k)$ encapsulates the cumulative effect of the original data point's removal as it propagates through the entire training process. Notably, similar product terms appear frequently in related domains, including continual learning and deep learning theory (Zou et al., 2021; Evron et al., 2022; Wu et al., 2022; 2024; Ding et al., 2024).

### 3.2   DATA VALUE EMBEDDING

Building on (1), we extract the test-data-independent components and define *"data value embedding"* for a training point $z^* \in \mathcal{B}_{t_s}$ as

$$\texttt{DVEmb}^{(t_s)}(z^*) := \eta_{t_s} \left[ \prod_{k=t_s+1}^{T-1} (\boldsymbol{I} - \eta_k \mathbf{H}_k) \right] \nabla \ell(\theta_{t_s}, z^*) \quad (2)$$

This embedding encapsulates the cumulative effect of a training point across the entire learning trajectory. By precomputing and storing these data value embeddings during or after the training phase, we enable highly efficient computation of data influence scores. Specifically, for any given test point $z^{(val)}$, the influence of a training point $z^*$ can be quickly determined by simply computing the dot product $\nabla \ell(\theta_T, z^{(val)})^\top \texttt{DVEmb}^{(t_s)}(z^*)$. Vector dot products are among the most computationally efficient operations, especially when executed on modern GPU hardware, which is optimized for such parallelized vector operations. Precomputing the data value embeddings eliminates the need for costly retraining or the availability of test data in advance, making the computation of data influence nearly instantaneous. This is particularly advantageous in real-world scenarios such as data marketplaces, where rapid, on-demand data attribution is critical.

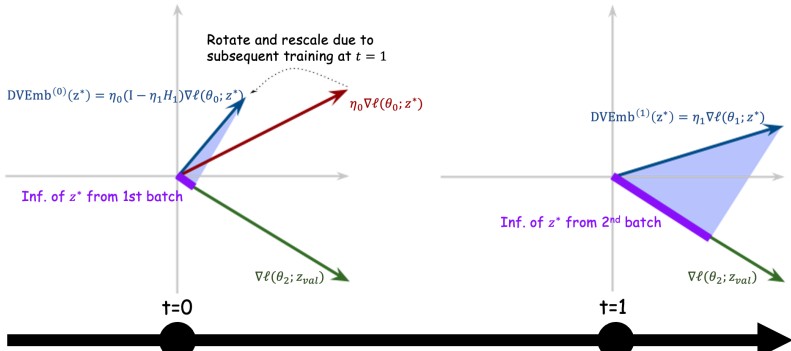

Figure 2: An illustrative example of data value embedding for a 2-step training. The influence of a training point $z^*$ on a test point $z^{(val)}$ can be obtained by projecting its data value embedding on $z^{(val)}$'s gradient vector at the final checkpoint.

**Approximation Error Bound.** In Appendix C.2, we derive a new theoretical analysis of the approximation error associated with the unrolled differentiation estimator for non-convex loss functions. We demonstrate that when the learning rate schedule satisfies $\eta_t \in \mathcal{O}(1/\sqrt{t})$ with the maximum learning rate scaling as $\mathcal{O}(1/\sqrt{T})$—a common choice in the literature (Vaswani, 2017)—the approximation error remains uniformly bounded and is *independent* of the total number of training steps $T$. While the proof relies on certain assumptions to abstract the complexities of real-world implementation, the theoretical result still implies the method's applicability in practical model training.

## 4 EFFICIENT COMPUTATION AND STORAGE OF DATA VALUE EMBEDDING

While the data value embedding approach offers a promising solution for real-time data attribution that incorporates training-specific factors, its practical implementation faces significant computational and storage challenges. The computation of `DVEmb` is non-trivial, requiring per-sample gradient calculations and per-step Hessian computations. Moreover, each `DVEmb`$_t(z^*)$ has the same dimensionality as the model parameters, making it infeasible to store individual embeddings for each training data point on the disk. To address these challenges, we develop a series of techniques that significantly enhance both the computational and storage efficiency of data value embedding.

### 4.1 RECURSIVE APPROXIMATION OF DATA VALUE EMBEDDING VIA GENERALIZED GAUSS-NEWTON MATRIX

We show that data value embedding can be computed recursively, beginning from the final training iteration and working backward, when using the Generalized Gauss-Newton (GGN) approximation for the Hessian matrix. This naturally gives rise to a backward computation algorithm for `DVEmb`$^{(t)}$.

A widely-adopted approximation for the Hessian matrix $\mathbf{H}_k$ is the Generalized Gauss-Newton (GGN) approximation $\mathbf{H}_t \approx \sum_{z \in \mathcal{B}_t} \nabla\ell(\theta_t, z)\nabla\ell(\theta_t, z)^\top$, particularly in the context of cross-entropy loss (Martens, 2020). The GGN approximation is extensively used in various machine learning algorithms because it captures the essential curvature information of the loss landscape while remaining computationally feasible. For further details, see Appendix C.4. Under this approximation to $\mathbf{H}_t$, the following shows that we can compute `DVEmb`$^{(t_s)}(z^*)$ for any $z^* \in \mathcal{B}_{t_s}$ if the data value embeddings of data points from later training iterations (i.e., `DVEmb`$^{(t)}(z)$ for $t \geq t_s + 1$) is available.

**Theorem 2.** *Given generalized Gauss-Newton approximation $\mathbf{H}_t \approx \sum_{z \in \mathcal{B}_t} \nabla\ell(\theta_t, z)\nabla\ell(\theta_t, z)^\top$, we have*

$$DVEmb^{(t_s)}(z^*) = \eta_{t_s}\nabla\ell(\theta_{t_s}, z^*) - \eta_{t_s}\sum_{t=t_s+1}^{T-1}\left(\sum_{z \in \mathcal{B}_t}\left(\nabla\ell(\theta_t, z)^\top\nabla\ell(\theta_{t_s}, z^*)\right)DVEmb^{(t)}(z)\right)$$

The proof is deferred to Appendix C.3.

**Interpretation.** Theorem 2 provides crucial insights into the interactions between training data points throughout the model training process. When two points $z^*$ and $z$ are similar, their gradient similarity term $\nabla\ell(\theta_t, z)^\top \nabla\ell(\theta_{t_s}, z^*)$ increases, indicating stronger interaction between these points. To illustrate this phenomenon, consider training a language model where an early data point $z^*$ contains content about "quantum computing". The influence of $z^*$ on the final model varies depending on the subsequent training data: if multiple similar "quantum computing" data points appear in later iterations, $z^*$'s influence on the final model diminishes, as these later examples could teach similar concepts to the model. Conversely, if $z^*$ remains one of the few "quantum computing" examples throughout training, it maintains a stronger influence on the final model.

**Overview of the remaining sections.** Theorem 2 suggests the possibility of a backpropagation algorithm for computing data value embeddings, contingent on the availability of per-sample gradient vectors for all training data. To make this approach practical for large-scale applications, we address two key challenges in the following sections: (1) Efficient computation and storage of per-sample gradient vectors for all training data (Section 4.2). (2) Efficient computation (Sections 4.3) and parallelization (Section 4.4) of data value embeddings using Theorem 2. Additionally, we discuss practical extensions and considerations for real-world scenarios (Appendix C.10).

## 4.2 STEP 1: STORE PER-SAMPLE TRAINING GRADIENT INFORMATION AT EACH ITERATION

During model training, we additionally store the *per-sample* gradient for each data point in the training batch. However, this approach presents significant computational and storage challenges: **(1) Storage:** Let $p$ denote the number of model parameters. Each gradient vector has dimension $p$, requiring $\mathcal{O}(TBp)$ disk space, where $B = |\mathcal{B}_t|$ is the batch size. This effectively corresponds to storing millions of model-size vectors. **(2) Efficiency:** Computing per-sample gradients necessitates separate backpropagation for each $z \in \mathcal{B}_t$, increasing computational cost by a factor of $B$.

**Avoiding per-sample gradient computation & full gradient storage (detailed in Appendix C.5).** To mitigate both issues, we leverage a gradient decomposition and take advantage of the computations already performed during backpropagation (Wang et al., 2025; Choe et al., 2024). By expressing gradients as the outer product of activations and output derivatives, only a single backpropagation on the aggregated loss is required to compute per-sample gradients, preserving the usual training speed. Additionally, instead of storing the full gradient vectors, we store the decomposed components, potentially reducing the storage requirement to $\mathcal{O}(TB\sqrt{p})$ for non-sequential data.

**Random projections for large models.** For large-scale foundation models with billions of parameters, we apply random projections to further compress the stored gradient information. Using projection matrices, we project the activations and output derivatives to a lower-dimensional space. This approach significantly reduces storage needs to $\mathcal{O}(TB\tilde{p})$, where $\tilde{p}$ is the projected dimension, while still capturing essential gradient geometric information.

We acknowledge that deriving a theoretical multiplicative guarantee here is challenging, given that the data value embedding itself is a linear combination that could be zero. However, our ablation study in Appendix E.5 demonstrates that our approach is relatively more robust compared to influence functions across different projection dimensions. These results provide strong evidence of the robustness of our method in practice, and we leave the theoretical guarantee as future work.

## 4.3 STEP 2: BACKPROPAGATING DATA VALUE EMBEDDING

Having established the method for storing projected gradient vectors, we now proceed to describe the backward computation algorithm for data value embeddings. For ease of presentation, we continue to use full gradient vector notation. However, in practical implementations, we use the projected gradient vectors for efficient storage. That is, $\nabla_\theta \ell \in \mathbb{R}^{\tilde{p}}$ in the subsequent contents.

According to Theorem 2, an equivalent expression for $\mathrm{DVEmb}^{(t_s)}(z^*)$ is given by

$$\mathrm{DVEmb}^{(t_s)}(z^*) = \eta_{t_s}\nabla\ell(\theta_{t_s}, z^*) - \eta_{t_s}\nabla\ell(\theta_{t_s}, z^*)\mathbf{M}^{(t_s)}$$

where $\mathbf{M}^{(t_s)} := \sum_{t=t_s+1}^{T-1}\left(\sum_{z\in\mathcal{B}_t}\left(\mathrm{DVEmb}^{(t)}(z)\nabla\ell(\theta_t, z)^\top\right)\right)$. At a high level, our algorithm computes $\mathrm{DVEmb}^{(t_s)}(z^*)$ for each $t_s$ from $T-1$ down to $0$, while maintaining a running matrix $\mathbf{M}^{(t_s)} \in \mathbb{R}^{\tilde{p}\times\tilde{p}}$ throughout the backpropagation process for algorithm efficiency.

**Backward algorithm from the final iteration.** We initialize $\mathbf{M}^{(T-1)} = \mathbf{0}$ as the data value embedding coincides with the training gradient for the last iteration. For $t_s = T - 1, \ldots, 0$, we recursively compute: **(1)** The data value embedding for each $z^* \in \mathcal{B}_{t_s}$: $\mathtt{DVEmb}^{(t_s)}(z^*) = \eta_{t_s} \nabla \ell(\theta_{t_s}, z^*) - \eta_{t_s} \mathbf{M}^{(t_s)} \nabla \ell(\theta_{t_s}, z^*)$, and **(2)** Update the weighting matrix after computing all embeddings for the current iteration: $\mathbf{M}^{(t_s-1)} = \mathbf{M}^{(t_s)} + \sum_{z^* \in \mathcal{B}_{t_s}} \mathtt{DVEmb}^{(t_s)}(z^*) \nabla \ell(\theta_{t_s}, z^*)^\top$. A detailed algorithm pseudocode can be found in Algorithm 1.

**Computing data value embedding on a per-layer basis.** Moreover, by adopting an assumption similar to that in EK-FAC regarding the independence of gradients across different layers, we can compute data value embeddings on a per-layer basis. This approach significantly reduces the computational and memory costs. The assumption of layer-wise independence is common in the literature on influence functions (Grosse et al., 2023), as it enables tractable analysis and efficient algorithms for deep neural networks. While this approximation neglects cross-layer gradient correlations, it is often justified because intra-layer interactions tend to dominate in practice. Treating layers independently thus strikes a favorable balance between computational feasibility and approximation accuracy.

**Complexity analysis. (1) Computational & Memory:** The primary computational cost of our algorithm stems from matrix multiplications and additions in updating data value embeddings and the weighting matrix, resulting in $\mathcal{O}(BT\tilde{p}^2)$ floating-point operations (flops). However, if we compute the data value embedding per layer, flops improve to $\mathcal{O}(BT\tilde{p}^2/L)$ where $L$ is the number of layers. The update of the running matrix $\mathbf{M}^{(t_s)}$ requires $\mathcal{O}(B\tilde{p}^2/L^2)$ memory. In comparison, regular model training requires $\mathcal{O}(BTp)$ flops and $\mathcal{O}(p)$ memory, where $p$ is the number of model parameters. Consequently, Algorithm 1 incurs significantly lower costs compared to regular training. We further note that the influence function method requires computing the per-sample gradient for each training data point on the final model, which is effectively equivalent to one epoch of training. As a result, both the memory requirements and flops for the influence function method are at least equivalent to those of model training, which are much larger than our algorithm's requirements. **(2) Storage:** Each $\mathtt{DVEmb}^{(t)}(z^*)$ has dimension $\mathcal{O}(\tilde{p})$, resulting in a total storage requirement of $\mathcal{O}(BT\tilde{p})$ for data value embeddings across all training points. While this can be substantial, disk storage is relatively inexpensive in modern computing environments. Moreover, the reduced dimensionality achieved through projection significantly mitigates the storage burden compared to storing full-dimensional embeddings. A summary of the complexity comparison with the most efficient implementation of the influence function (Choe et al., 2024) is provided in Table 2 in Appendix C.9.

## 4.4 PARALLELIZED EXTENSION FOR INFLUENCE EMBEDDING COMPUTATION (OVERVIEW)

The backpropagation algorithm introduced in Section 4.3 operates with a runtime complexity of $\mathcal{O}(T)$, as it sequentially computes $\mathtt{DVEmb}^{(t_s)}$ for $t_s = T - 1, \ldots, 0$. While being significantly more efficient than the influence function, which requires re-computing all training gradients on the final model (see Section 5.2 and Table 2), it can still be costly for long training periods. Here, we introduce *influence checkpointing*, a parallelized extension for Algorithm 1.

**Influence Checkpointing.** We reduce computational costs by allowing concurrent computation of data value embeddings at multiple checkpoints during training. By selecting $K$ evenly spaced training steps, we can efficiently compute data value embeddings for each *intermediate checkpoint* in parallel. By carefully computing and storing necessary results, we can efficiently reconstruct the data value embedding for the final model. This reduces the overall computational cost by $K$ times. The detailed algorithm description, pseudocode, and complexity analysis are deferred to Appendix C.7.

**Data Value Dynamics During Training.** In addition to its computational benefits, the influence checkpointing algorithm enables a powerful capability: tracking the evolution of data influences throughout the entire model training process. If the intermediate checkpoints $\theta_{t_1}, \ldots, \theta_{t_{K-1}}$ was saved—a common practice in foundation model pretraining—we can analyze how the influence of a fixed data point changes on different intermediate checkpoints. As a result, we gain a more fine-grained and dynamic view of how the influence of a fixed data point propagates to the subsequent training steps, providing deeper insights into the model's learning behavior over time. This capability opens up new avenues for understanding and optimizing machine learning model training.

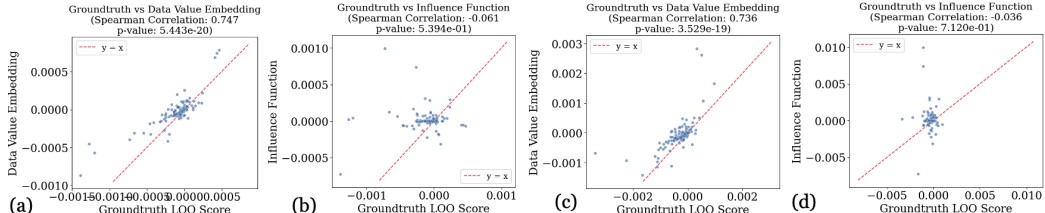

Figure 3: The correlation between ground-truth LOO when the MLP is trained for 3 epochs and the estimation obtained by (a) the data value embedding method and (b) the influence function for *single epoch removal*. (c) and (d) present the corresponding correlations for *all-epoch removal*. Additional results for models being trained for a longer time can be found in Appendix E.2.

## 5 EXPERIMENTS

In this section, we evaluate the effectiveness of our proposed data value embedding method. First, we assess its fidelity in accurately reflecting data importance using small-scale experimental setups (Section 5.1), as well as its computational efficiency (Section 5.2). We then apply data value embedding to analyze the training dynamics during foundation model pretraining (Section 5.3 and Appendix E.4). The implementation details and additional results are deferred to Appendix E.

### 5.1 FIDELITY EVALUATION

To validate the effectiveness of our proposed data value embedding algorithm, we assess its accuracy in approximating TSLOO scores. Additionally, in Appendix E.2.1, we compare to a variety of data attribution baselines on the standard benchmarks of mislabel data detection and data selection.

Computing ground-truth LOO requires retraining the model multiple times, each time excluding a single data point while keeping all other training specifics, such as batch order, unchanged. Given the computational intensity, we conduct our experiments on the MNIST (LeCun et al., 1989) using a small MLP trained with standard SGD. We consider two settings: **(1) Single epoch removal**, where a data point is excluded from training during a single epoch but still in other training epochs. Here, we remove the data point from the last epoch. **(2) All-epoch removal**, where a data point is excluded in all epochs. In this case, the approximation provided by data value embedding is obtained by summing the data value embeddings of the data point from all epochs, as discussed in Appendix C.10.

Figure 3 shows that data value embedding has a high Spearman correlation with the ground-truth LOO. This superior performance is consistent across both settings. We note that the influence function scores remain constant for both settings, as influence functions do not account for specific training runs and cannot differentiate between single- and multi-epoch removals. Moreover, influence function exhibits a very weak correlation with LOO, a phenomenon that has been reported in many literature (Søgaard et al., 2021; Basu et al., 2020; Bae et al., 2022; Epifano et al., 2023).

### 5.2 COMPUTATIONAL EFFICIENCY

In this section, we compare the storage, memory, and computational efficiency of data value embedding with LoGRA (Choe et al., 2024), the most efficient implementation of the influence function so far. LoGRA first computes per-sample training gradients on the final model for *all* training data points $z^* \in \mathcal{D}$, where $\mathcal{D}$ represents the dataset. Like our algorithm, LoGRA also uses random projection and stores the *projected* Hessian-adjusted gradient $\mathbf{H}_T^{-1} \nabla \ell(\theta_T, z^*)$ to the disk, and the influence function can be computed via dot-product with test data gradient.

Table 1 shows the result of computing data influence for Pythia-410M trained on 1% of the Pile dataset. Both algorithms first compute and store Hessian-adjusted gradients/data value embedding, and then compute the data influence with respect to any given test point. As we can see, LoGRA and data value embedding have similar disk storage requirements, as both approaches save vectors of dimension $\tilde{p}$ for each data point. For peak GPU memory in the storage step, LoGRA requires recomputing gradients for all training data on the final model $\theta_T$, which is effectively equivalent to one epoch of model training. In contrast, the data value embedding computation algorithm operates

only on projected vectors, which takes much less GPU memory (0.84 vs 63.6GB). Consequently, the computational efficiency for computing data value embeddings is also much higher (over $15\times$ faster). When computing data influence, since both approaches simply take the dot product between test data's (projected) gradient and $\mathbf{H}_T^{-1}\nabla\ell(\theta_T, z^*)$ or $\texttt{DVEmb}^{(t)}(z^*)$ or data value embedding, the GPU memory usage and efficiency are the same.

| | Storing $\mathbf{H}_T^{-1}\nabla\ell(\theta_T, z^*)$ / data value embedding | | | Compute Influence (dot-product) | |
| --- | --- | --- | --- | --- | --- |
| | **Storage** | **Peak GPU Mem.** | **Throughput** | **Peak GPU Mem.** | **Throughput** |
| **LoRGA** | 170GB | 63.6GB | 41.6 | 16.31GB | 640 |
| **Data Value Embedding** | 171GB | 64.6GB / 0.84GB* | 667.52 | 16.31GB | 640 |

Table 1: Memory and compute efficiency analysis for LoRGA (Choe et al., 2024) and data value embedding. Throughput is measured as the number of data points per second for storing and influence computation. The experiment is conducted on one A100 GPU with 80GB VRAM. The projection dimension is set to 1024. *Since data value embedding technique contains two different steps in storing relevant information for data attribution (storing gradient during training & compute and store data value embedding after training), we include the peak GPU memory usage for both steps.

## 5.3 ANALYZING TRAINING DYNAMICS OF FOUNDATION MODELS

In this section, we showcase data value embedding as a powerful tool for analyzing the training dynamics of foundation model pretraining with Pythia-410M trained on 1% of Pile dataset as an example. Results for additional datasets/models and the analysis for fine-tuning are in Appendix E.3.

**Value of training data from different stages in LLM pretraining.** We first visualize the distribution of data influence scores on the final model across different training batches. For a fair comparison, we normalize the influence scores for each batch by their learning rate. Figure 1 (a) illustrates the results for training Pythia-410M on the Pile dataset. As we can see, the data influence on the final model can be categorized into three distinct regimes: **(1) High-impact Warmup Phase:** This phase occurs during the very early training stage and is characterized by exceptionally high data influence scores. It corresponds to a brief window at the onset of training where the loss reduces rapidly. **(2) Low-impact Basin:** This regime spans the early-to-middle training stage, where data influence scores are significantly lower. This period coincides with a slowdown in the rate of loss decrease, transitioning into a phase of relative stability. **(3) Gradual Ascent:** In this phase, we observe that the later a data point participates in the training, the higher its influence score becomes.

**Explanation: (1) Parameter initialization and warmup training are important for final model performance.** During the very early stages of training, the gradient norms are large, which leads to significant parameter updates. Furthermore, the subsequent gradients' magnitude *decrease rapidly*, causing data points from the High-impact Warmup Phase to maintain substantial influence throughout the training process, even as their immediate impact diminishes over time.

Figure 4 visualizes this phenomenon. The purple curve shows that training data points from the High-impact Warmup Phase, while experiencing large drops in influence as training progresses, still maintain higher influence than later data points. This observation aligns with the well-known effect that model initialization and/or warm-up training plays a crucial role in training performance (He et al., 2015; Hanin & Rolnick, 2018), effectively initializing model parameters and gradually preparing the model for more complex learning tasks. **(2) Influence saturation from future data.** As training progresses into a smoother loss regime, the gradient norms become relatively stable and decrease slowly. This makes the influence decay from subsequent training much more significant for these data points compared to those from the High-Impact Warmup Phase. Since earlier data points experience more

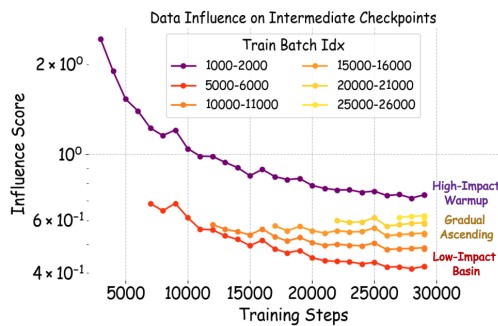

Figure 4: Evolution of influence scores across training checkpoints. The x-axis shows training iterations, and the y-axis shows the average influence of training examples on each checkpoint. Examples are grouped according to the iterations they are being trained on.

future training iterations, their influence decreases more over time. The red curve in Figure 4 demon-

strates this trend, showing influence scores for these points gradually decreasing during training and eventually falling below those of later training data points. One might initially think this phenomenon is connected to catastrophic forgetting, where the model appears to "forget" the influence of data from earlier training phases as it progresses. However, we note that a data point's influence score decreases the most when future data points are similar to it, which is different from catastrophic forgetting. Intuitively, if future points are identical, the presence of the earlier data point in training becomes less relevant to the model's behavior. A more detailed explanation is deferred to Appendix E.3.

**Implications for data selection strategies.** These observations suggest that for pretraining, data selection is most critical during the very early and later stages of training. To validate this insight, we train Pythia-410M on Pile with different online data selection strategies, as shown in Figure 1 (b). Specifically, we use an online data selection strategy (adapted from Fan et al. (2024)) that forms each training batch by selecting data points whose gradients align well with those from a validation batch sampled from Pile (see Appendix E.3.2 for details). This selection process requires computing gradient similarities, introducing significant overhead at each iteration where it is applied. Therefore, identifying the most critical training phases for applying this selection process becomes crucial for computational efficiency. Remarkably, Figure 1 (b) demonstrates that the performance of a strategy where we only perform data selection in the first 2000 iterations and after 20000 iterations closely matches the performance when data selection is performed in all iterations. Moreover, it reduces computational costs by more than 5 times. This corroborates our practical insights for designing efficient data selection strategies in LLM pretraining: by focusing data selection efforts on the critical early and late stages of training, we can potentially achieve optimal model performance while significantly reducing computational overhead.

# 6 CONCLUSION AND LIMITATIONS

In this paper, we introduced Data Value Embedding, a novel approach to data attribution tailored for foundation models. Our method addresses critical limitations of existing techniques by capturing the temporal dynamics of training and enabling real-time attribution without the need for model retraining. The experiments demonstrate the efficacy of data value embedding in providing accurate and efficient data influence scores and unveiling unique insights into the training dynamics of foundation models.

**SGD as a proxy for Adam.** The data value embedding in (2) is specifically tailored for SGD. It is not directly extendable to other popular optimizers like Adam due to their normalization terms. Nonetheless, using SGD as a proxy for Adam allows for efficient data influence estimation, which is the approach that is usually adopted in practice and has proved to be effective in our experiment, providing a practical and effective solution for the current scope of our work. While using as a proxy for Adam has proved to be effective in our experiment, extending data value embedding to Adam and other optimizers remains an exciting direction for future research.

**Training curriculum design.** Our findings on the varying influence of data points across training stages suggest the potential for designing optimal training curricula. Future work could explore leveraging data value embedding to design curricula that maximize learning efficiency. This could involve dynamically adjusting the presentation order and frequency of data points based on their predicted influence at different training stages.

ACKNOWLEDGMENT

This work is supported in part by the National Science Foundation under grants IIS-2312794, IIS-2313130, OAC-2239622, CNS-2131938, CNS-2424127, Amazon-Virginia Tech Initiative in Efficient and Robust Machine Learning, the Commonwealth Cyber Initiative, Cisco, OpenAI and Google.

We thank Tong Wu, Meng Ding, Haizhou Shi, and Weida Li for their helpful feedback on the preliminary version of this work.

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

## A    EXTENDED RELATED WORKS

Here, we provide a general overview of the literature that is relevant to the trajectory-specific LOO. We refer the readers to Appendix A of Wang et al. (2025) for the overview of the related works for Data Shapley and other data attribution techniques.

### A.1    LOO INFLUENCE VS LOOCV

It is important to distinguish our LOO influence measure from traditional Leave-One-Out Cross-Validation (LOOCV) (Ziegel, 2003). While both involve removing individual data points, they serve different purposes and yield different interpretations. LOOCV is a model evaluation technique that estimates generalization performance by averaging prediction errors on held-out examples, where smaller errors indicate better model performance. In contrast, LOO influence measures how removing a specific training point affects the model's behavior on validation data, quantifying each training example's importance to the learning process. While LOOCV requires training $N$ separate models to evaluate generalization (where $N$ is the dataset size), LOO influence focuses on understanding the counterfactual impact of individual training points on model behavior. This distinction is crucial as we aim to understand data importance rather than model performance.

### A.2    INFLUENCE FUNCTION AND FRIENDS

Influence function (Koh & Liang, 2017) has emerged as an important tool for interpreting and analyzing machine learning models. As the influence function requires computing the Hessian inverse, many subsequent works are focusing on improving the scalability of the influence function for large-scale models (Guo et al., 2021; Schioppa et al., 2022; Grosse et al., 2023). More recently, Kwon et al. (2023) developed an efficient influence function approximation algorithm that is suitable for LoRA fine-tuning, and Zhang et al. (2024) extends the influence function to time-series datasets. In a similar spirit to us, Chen et al. (2020) a multi-stage extension of influence function to trace a fine-tuned model's behavior back to the pretraining data. However, they changed the original loss function and added a regularization term to account for intermediate checkpoints. The most closely related to our work is Choe et al. (2024). Similar to us, they also make use of the low-rank gradient decomposition and random projection to enable efficient computation and storage of per-sample gradient. However, their approach still requires computing per-sample gradient vectors for all training data on the final model checkpoint, which is effectively equivalent to one model retraining and takes a significantly longer time than data value embedding.

**Influence function and Newton step.** The influence function formula bears a striking resemblance to the Newton step in optimization, as both involve the product of an inverse Hessian and a gradient. As a result, much data attribution literature simply describes the influence function as "one Newton step". There is a subtle difference in their expressions and motivations. For large dataset, one Newton step approximates the influence function under certain regularity conditions. A single Newton step from the full model parameters toward the leave-one-out solution was proposed as an influence measure by Pregibon (1981) and has since been leveraged for data attribution (Park et al., 2023). The high-level argument is that while obtaining the exact leave-one-out model requires fully minimizing the leave-one-out loss, taking one Newton step from the optimal model $\theta^*$ trained on the full dataset often provides sufficient information to distinguish the relative influence of different data points.

## B  Limitations of the Existing Data Attribution Techniques for Foundation Models

### B.1  Influence Function

Influence functions (Cook & Weisberg, 1980; Koh & Liang, 2017) are a classical technique from robust statistics, adapted for machine learning to measure how the removal of a single data point affects the performance of a trained model. Influence functions quantify the sensitivity of a model's predictions to specific data points, offering insights into the importance of individual training samples. In the machine learning framework, they are particularly useful for diagnosing model behavior, understanding dataset quality, and identifying mislabeled or harmful data points. The core idea of influence functions is to approximate the effect of removing a data point from the training set without needing to retrain the model. Instead of actually excluding a point and retraining, influence functions leverage the model's final parameters and compute the impact of a point's removal based on the gradient and Hessian inverse of the loss function at the final model state. Formally, the influence of a training data point $z_i$ on the loss $\ell(\theta, z^{(val)})$ at a validation point $z^{(val)}$ is defined as:

$$\mathbf{IF}(z_i) := -\nabla_\theta \ell(\theta, z^{(val)})^\top \mathbf{H}^{-1} \nabla_\theta \ell(\theta, z_i)$$

where $\theta$ is the final model parameter after training, $\mathbf{H} = \frac{1}{N} \sum_{i=1}^{N} \nabla_\theta^2 \ell(\theta, z_i)$ is the Hessian of the total training loss at $\theta$, $\nabla_\theta \ell(\theta, z^{(val)})$ and $\nabla_\theta \ell(\theta, z_i)$ are the gradients of the loss at the validation point and the training point, respectively.

**Limitation: Neglecting Training Phases and Unrealiable Approximation to LOO.** A key limitation of influence function techniques is their exclusive focus on the final model parameters, thereby ignoring the intermediate dynamics of the training process. By assessing data contributions solely based on the final trained model, influence functions fail to capture how each data point influenced the model's updates throughout training. This narrow focus introduces inaccuracies, as it overlooks the cumulative effects of model fluctuations during the training iterations. Consequently, influence functions can be less accurate in evaluating data contributions, particularly in large-scale models where the training process plays a significant role. For instance, in modern training paradigms for large language models (LLMs), models are typically pretrained on a broad corpus and subsequently fine-tuned on specialized domains. Influence functions, however, cannot differentiate between the impacts of data points during pretraining and fine-tuning phases. Relying solely on the final model parameters after fine-tuning, they miss how pretraining data contributed to learning general language structures or how fine-tuning data adapted the model to specific domains. This inability to account for different training stages results in incomplete and often noisy estimates of data contributions, thereby reducing the precision of attribution in multi-stage training processes.

Moreover, our analysis in Section D demonstrates that the influence function approximates the expected data influence across different training trajectories only under overly simplistic conditions, which are often violated in practice. These conditions, such as assuming identical intermediate model checkpoints and Hessian matrices, almost never hold in real-world training scenarios where model evolve significantly. This highlights the inadequacy of influence functions in accurately capturing data contributions, underscoring the necessity for more comprehensive data attribution methods that consider the entire training trajectory.

**Neglecting Training Phases Necessitates Unreasonable Assumptions and Often Require Model Retraining.** Additionally, the focus on the final model necessitates assumptions of convergence and strong convexity to ensure reliable results. In many real-world settings, where models are non-convex and may not fully converge, these assumptions are often violated, leading to further inaccuracies in the data contribution estimates. As the influence function score is often found to be highly noisy in practice (Basu et al., 2020; Søgaard et al., 2021; Bae et al., 2022; Epifano et al., 2023), it typically necessitates multiple model retraining to produce reasonable results (Deng et al., 2024), which can undermine their original computational efficiency advantage.

### B.2  In-Run Data Shapley

In-Run Data Shapley (Wang et al., 2025) is a data attribution technique designed to evaluate the contribution of individual data points during a single training run of machine learning models. It

builds on the traditional Data Shapley framework, which stems from cooperative game theory. The Shapley value, originally proposed by Lloyd Shapley in 1953, distributes total utility fairly among all contributing players based on their marginal contributions. Applying this concept to machine learning, Data Shapley attributes the contribution of each data point in a training dataset by assessing its influence on model performance. However, standard Data Shapley methods face limitations in scalability because they require numerous retraining iterations on different data subsets. These computational demands make them impractical for large-scale models such as foundation models. To address these challenges, In-Run Data Shapley was introduced as a scalable alternative that avoids the need for repeated retraining. Instead, it leverages the iterative nature of model training, specifically neural networks, where parameters are updated in small increments. By tracking gradient updates at each training step, In-Run Data Shapley calculates the contribution of individual data points toward the final model without retraining. It approximates the Shapley value using local utility functions tied to specific gradient updates and extends these to the full training process, capturing cumulative contributions. This method reduces the computational overhead to a level comparable with standard training runs while maintaining the theoretical fairness and interpretability of Shapley values.

**Limitation: Requirement of Validation Data in Advance.** One of the key limitations of In-Run Data Shapley is its reliance on the availability of validation data prior to the start of training. The technique calculates data contribution by examining the impact of training points on model performance as measured against the validation set. Thus, access to this validation data throughout the training process is necessary to compute meaningful Shapley values at each iteration. This restriction can limit the applicability of In-Run Data Shapley in scenarios where validation data is not immediately available, such as in certain real-time learning environments or when the validation set is defined only after training. Potential workarounds, such as saving intermediate model checkpoints to calculate contributions post-training, add complexity to the process and might be unreliable.

# C  ALGORITHM DETAILS

## C.1  DERIVATION DETAILS FOR SECTION 3.1

Suppose $z^*$ is a data point that participates in the training during the *first* iteration. Denote $\mathcal{B}_t$ as the training batch in the $t$-th iteration. For standard Stochastic Gradient Descent (SGD), we have:

$$\theta_{k+1} = \theta_k - \eta_k \sum_{z \in \mathcal{B}_k} \nabla \ell(\theta_k, z) \tag{3}$$

for $k = 0, \ldots, T-1$, where $\eta_k$ is the learning rate at iteration $k$.

For validation data $z^{(val)}$, we aim to estimate the change in $\ell(\theta_T, z^{(val)})$ by removing $z^*$ from the first iteration. Specifically, we want to estimate $\ell(\theta'_T, z^{(val)}) - \ell(\theta_T, z^{(val)})$ where:

$$\theta'_1 = \theta_0 - \eta_0 \sum_{z \in \mathcal{B}_0 \setminus \{z^*\}} \nabla \ell(\theta_0, z) \tag{4}$$

and

$$\theta'_{k+1} = \theta'_k - \eta_k \sum_{z \in \mathcal{B}_k} \nabla \ell(\theta'_k, z) \tag{5}$$

for $k = 1, \ldots, T-1$.

To approach this problem, we define an interpolation between $\theta_T$ and $\theta'_T$:

$$\theta_1(\varepsilon) := \theta_0 - \eta_0 \sum_{z \in \mathcal{B}_0 \setminus \{z^*\}} \nabla \ell(\theta_0, z) - \eta_0(1 - \varepsilon) \nabla \ell(\theta_0, z^*) \tag{6}$$

where $\theta_T(\varepsilon)$ is defined accordingly. Note that $\theta_T(0) = \theta_T$ and $\theta_T(1) = \theta'_T$.

By taking the first-order Taylor expansion at $\varepsilon = 0$, we have:

$$\ell(\theta'_T, z^{(val)}) - \ell(\theta_T, z^{(val)}) = \ell(\theta_T(1), z^{(val)}) - \ell(\theta_T(0), z^{(val)})$$

$$\approx \frac{\partial}{\partial \varepsilon} \ell(\theta_T(\varepsilon), z^{(val)}) \Big|_{\varepsilon=0}$$

$$= \nabla \ell(\theta_T, z^{(val)})^\top \frac{\partial \theta_T(\varepsilon)}{\partial \varepsilon} \Big|_{\varepsilon=0} \tag{7}$$

Now, we derive $\frac{\partial \theta_T(\varepsilon)}{\partial \varepsilon}\Big|_{\varepsilon=0}$ by observing the following recursive relation for all $k \geq 1$:

$$\frac{\partial \theta_{k+1}(\varepsilon)}{\partial \varepsilon} = \frac{\partial \theta_k(\varepsilon)}{\partial \varepsilon} - \eta_k \sum_{z \in \mathcal{B}_k} \nabla^2 \ell(\theta_k(\varepsilon), z) \frac{\partial \theta_k(\varepsilon)}{\partial \varepsilon} \tag{8}$$

$$= \frac{\partial \theta_k(\varepsilon)}{\partial \varepsilon} (\boldsymbol{I} - \eta_k \mathbf{H}_k(\varepsilon)) \tag{9}$$

where $\mathbf{H}_k(\varepsilon) = \sum_{z \in \mathcal{B}_k} \nabla^2 \ell(\theta_k(\varepsilon), z)$ is the Hessian and $\boldsymbol{I}$ is the identity matrix. Additionally, for the first iteration where $z^*$ participates, we have

$$\frac{\partial \theta_1(\varepsilon)}{\partial \varepsilon} = \eta_0 \nabla \ell(\theta_0, z^*) \tag{10}$$

Expanding the recursion and substituting it back into our original expression, we get:

$$\ell(\theta_T(1), z^{(val)}) - \ell(\theta_T(0), z^{(val)}) \approx \frac{\partial}{\partial \varepsilon} \ell(\theta_T(\varepsilon), z^{(val)})|_{\varepsilon=0}$$

$$= \eta_0 \nabla \ell(\theta_T, z^{(val)})^\top \underbrace{\left[ \prod_{k=1}^{T-1} (\boldsymbol{I} - \eta_k \mathbf{H}_k) \right]}_{\text{cumulative effect}} \nabla \ell(\theta_0, z^*)$$

This final expression gives an estimate of the influence of removing $z^*$ from the first iteration on the loss on $z^{(val)}$ at the final iteration. The term $\prod_{k=1}^{T-1} (\boldsymbol{I} - \eta_k \mathbf{H}_k)$ represents the *cumulative effect of all training iterations on the initial influence.* This product captures how the impact of the initial change propagates through the entire training process, accounting for the learning rate and the training data at each subsequent step.

C.2   Error Guarantee for Unrolling-based Approach

In this section, we derive the approximation error guarantee of the unrolling differentiation estimator

$$\Delta\theta_{-z^*} := \frac{\partial\theta_T(\varepsilon)}{\partial\varepsilon}\bigg|\varepsilon = 0 = \eta t_s \left[\prod_{k=t_s+1}^{T-1}(\boldsymbol{I} - \eta_k\mathbf{H}_k)\right]\nabla\ell(\theta_{t_s}, z^*),$$

for non-convex loss functions. A very loose bound for $\|\theta_T - \theta'_T - \Delta\theta_{-z^*}\|$ has been derived in Hara et al. (2019). Here, we improve the error bound by additionally considering the decay of the learning rate and the spectral norm of Hessian matrices as training progresses. Notably, we establish a uniform bound on the gap.

Assume that $\ell(z; \theta)$ is twice differentiable with respect to the parameter $\theta$, and we train the model for $T$ iterations. We make the following assumptions:

1. **Learning Rate Schedule:** The learning rate $\eta_t$ at iteration $t$ follows the schedule $\eta_t = \frac{\eta_{\max}}{\sqrt{t}}$ where $\eta_{\max} = \frac{C}{\sqrt{T}}$ for some constant $C$. **Justification:** The decaying learning rate schedule $\eta_t = \frac{\eta_{\max}}{\sqrt{t}}$ is a common choice in neural network training in famous literature (Vaswani, 2017). This schedule allows for larger step sizes during the initial phases of training, facilitating rapid convergence, while gradually reducing the step sizes to fine-tune the model parameters and ensure stability as training progresses. The max learning rate $\eta_{\max} = \mathcal{O}\left(\frac{1}{\sqrt{T}}\right)$ ensures that the cumulative step sizes remain bounded over $T$ iterations, which is crucial for deriving meaningful error bounds. This approach balances the trade-off between exploration and convergence, making it well-suited for training deep neural networks where maintaining stability is essential.

2. **Hessian Spectral Norm Decay:** There exists a constant $\Lambda > 0$ such that the Hessian matrices satisfy $\mathbf{H}_t \preceq \frac{\Lambda}{\sqrt{t}}\boldsymbol{I}$ for all $t \geq 1$. **Justification:** The assumption that the spectral norm of the Hessian matrices decays as $\mathbf{H}_t \preceq \frac{\Lambda}{\sqrt{t}}\boldsymbol{I}$ is grounded in the observation that, as training progresses, the optimization landscape often becomes flatter around minima. This reduction in curvature implies that the Hessian's eigenvalues decrease, leading to smaller spectral norms. Such behavior is typical in many deep learning scenarios where initial training steps navigate regions of high curvature, followed by stabilization in flatter regions as the model converges. Additionally, this assumption aligns with empirical findings in deep learning literature (), where Hessian's spectral norm has been observed to decrease over time, thereby facilitating more stable and efficient convergence. By incorporating this decay, we account for the diminishing influence of curvature on parameter updates, which is critical for tightening the error bounds in our analysis.

Under these assumptions, we proceed to derive a uniform bound on the approximation error $\|\theta_T - \theta'_T - \Delta\theta_{-z^*}\|$. This bound provides theoretical guarantees for the effectiveness of the unrolling-based approach in estimating the influence of removing a training data point on the final model parameters. The derivation leverages the decaying learning rate and the diminishing spectral norm of the Hessian matrices to tighten the error bounds compared to previous work (Hara et al., 2019).

**Theorem 3.** *Assume that $\ell(z; \theta)$ is twice differentiable, that the Hessian $\nabla_\theta^2\ell(z; \theta)$ is L-Lipschitz continuous with respect to $\theta$, and that the gradient norm is bounded, i.e., $\|\nabla_\theta\ell(z; \theta)\| \leq G$ for all $z$ and $\theta$. Furthermore, assume that the learning rate $\eta_t$ at iteration $t$ follows the schedule $\eta_t = \frac{\eta_{\max}}{\sqrt{t}}$, where $\eta_{\max} = \frac{C}{\sqrt{T}}$ for some constant $C > 0$. Then, for the unrolling differentiation estimator $\Delta\theta_{-z^*}$, the approximation error satisfies*

$$\|(\theta_T - \theta'_T) - \Delta\theta_{-z^*}\| \leq \frac{32}{3}G^2C^3Le^{C\Lambda} \tag{11}$$

*Proof.* By Cauchy's Mean Value Theorem, for each iteration $s \in \{t_s, \ldots, T-1\}$, there exists $r \in [0, 1]$ such that for $\theta_s^* := r\theta'_s + (1-r)\theta_s$, we have

$$\sum_{z\in\mathcal{B}_s}(\nabla_\theta\ell(z; \theta'_s) - \nabla_\theta\ell(z; \theta_s)) = \mathbf{H}_s^*(\theta'_s - \theta_s),$$

where $\mathbf{H}_s^* := \sum_{z \in \mathcal{B}_s} \nabla_\theta^2 \ell(z; \theta_s^*)$. Define $Z_s := (\boldsymbol{I} - \eta_s \mathbf{H}_s)$ and $Z_s^* := (\boldsymbol{I} - \eta_s \mathbf{H}_s^*)$. Then, we have

$$\theta'_{s+1} - \theta_{s+1} = Z_s(\theta'_s - \theta_s) + \eta_s(\mathbf{H}_s - \mathbf{H}_s^*)(\theta'_s - \theta_s) = Z_s(\theta'_s - \theta_s) + D_s,$$

where $D_s := \eta_s(\mathbf{H}_s - \mathbf{H}_s^*)(\theta'_s - \theta_s)$. Recursively applying these equalities over $s \in \{t_s, \dots, T-1\}$, we obtain

$$\theta'_T - \theta_T = \Delta\theta_{-z^*} + \sum_{s=t_s}^{T-1} \prod_{k=s+1}^{T-1} Z_k D_s.$$

Hence, the approximation error is given by

$$\|(\theta_T - \theta'_T) - \Delta\theta_{-z^*}\| = \left\|\sum_{s=t_s}^{T-1} \prod_{k=s+1}^{T-1} Z_k D_s\right\|.$$

To bound this, we proceed as follows. Given the learning rate schedule $\eta_t = \frac{\eta_{\max}}{\sqrt{t}} = \frac{C}{\sqrt{Tt}}$, and the assumption that $\mathbf{H}_t \preceq \frac{\Lambda}{\sqrt{t}}\boldsymbol{I}$, we have

$$\|Z_k\| = \|\boldsymbol{I} - \eta_k \mathbf{H}_k\| \leq 1 + \eta_k \frac{\Lambda}{\sqrt{k}} = 1 + \frac{C\Lambda}{k\sqrt{T}}.$$

For large $T$ and $k \geq s \geq t_s \geq 1$, the term $\frac{C\Lambda}{k\sqrt{T}}$ is small. Thus, we can bound the product of the norms as

$$\prod_{k=s+1}^{T-1} \|Z_k\| \leq \exp\left(\sum_{k=s+1}^{T-1} \frac{C\Lambda}{k\sqrt{T}}\right) \leq \exp\left(\frac{C\Lambda}{\sqrt{T}} \sum_{k=s+1}^{T-1} \frac{1}{k}\right).$$

Using the harmonic series approximation,

$$\sum_{k=s+1}^{T-1} \frac{1}{k} \leq \ln\left(\frac{T}{s}\right) \leq \ln(T).$$

Thus,

$$\prod_{k=s+1}^{T-1} \|Z_k\| \leq \exp\left(\frac{C\Lambda \ln T}{\sqrt{T}}\right) \leq e^{C\Lambda}.$$

Therefore, we have

$$\left\|\sum_{s=t_s}^{T-1} \prod_{k=s+1}^{T-1} Z_k D_s\right\| \leq e^{C\Lambda} \sum_{s=t_s}^{T-1} \|D_s\|.$$

Next, we bound $\|D_s\|$:

$$\|D_s\| = \|\eta_s(\mathbf{H}_s - \mathbf{H}_s^*)(\theta'_s - \theta_s)\| \leq \eta_s \|\mathbf{H}_s - \mathbf{H}_s^*\| \cdot \|\theta'_s - \theta_s\|.$$

Since $\nabla_\theta^2 \ell(z; \theta)$ is $L$-Lipschitz continuous with respect to $\theta$, we have

$$\|\mathbf{H}_s - \mathbf{H}_s^*\| \leq L\|\theta'_s - \theta_s\|.$$

Additionally, we have

$$\|\theta'_s - \theta_s\| \leq 2 \sum_{t=1}^{s} \eta_t G = 2G \sum_{t=1}^{s} \frac{C}{\sqrt{Tt}} \leq 4GC \frac{\sqrt{s}}{\sqrt{T}},$$

where we used the bound $\sum_{t=1}^{s} \frac{1}{\sqrt{t}} \leq 2\sqrt{s}$.

Thus,

$$\|D_s\| \leq \eta_s L \cdot \left(4GC \frac{\sqrt{s}}{\sqrt{T}}\right)^2 = \Gamma \frac{\sqrt{s}}{T^{1.5}}$$

where $\Gamma = 16G^2 C^3 L$.

Substituting this bound into the sum, we obtain

$$\left\| \sum_{s=t_s}^{T-1} \prod_{k=s+1}^{T-1} Z_k D_s \right\| \leq e^{C\Lambda} \sum_{s=t_s}^{T-1} \Gamma \frac{\sqrt{s}}{T^{1.5}}.$$

We now evaluate the summation:

$$\sum_{s=t_s}^{T-1} \frac{\sqrt{s}}{T^{1.5}} \leq \frac{1}{T^{1.5}} \sum_{s=1}^{T} \sqrt{s} \leq \frac{1}{T^{1.5}} \cdot \frac{2}{3} T^{1.5} = \frac{2}{3},$$

where we used the bound $\sum_{s=1}^{T} \sqrt{s} \leq \frac{2}{3} T^{1.5}$.

Therefore,

$$\left\| \sum_{s=t_s}^{T-1} \prod_{k=s+1}^{T-1} Z_k D_s \right\| \leq e^{C\Lambda} \Gamma \cdot \frac{2}{3} = \frac{32}{3} G^2 C^3 L e^{C\Lambda}.$$

$\square$

## C.3 Computing Data Value Embedding Recursively

**Theorem 4** (Restate for Theorem 2). *Given generalized Gauss-Newton approximation* $\mathbf{H}_t \approx \sum_{z \in \mathcal{B}_t} \nabla\ell(\theta_t, z)\nabla\ell(\theta_t, z)^\top$, *we have*

$$\text{DVEmb}^{(t)}(z^*) \approx \eta_t \nabla\ell(\theta_t, z^*) - \eta_t \sum_{k=t+1}^{T-1} \left( \sum_{z \in \mathcal{B}_k} \left( \nabla\ell(\theta_k, z)^\top \nabla\ell(\theta_t, z^*) \right) \text{DVEmb}^{(k)}(z) \right)$$

*Proof.*

$\text{DVEmb}^{(t)}(z^*)$

$$= \eta_t \left[ \prod_{k=t+1}^{T-1} (\boldsymbol{I} - \eta_k \mathbf{H}_k) \right] \nabla\ell(\theta_t, z^*)$$

$$= \eta_t \left[ \prod_{k=t+2}^{T-1} (\boldsymbol{I} - \eta_k \mathbf{H}_k) \right] (\boldsymbol{I} - \eta_{t+1} \mathbf{H}_{t+1}) \nabla\ell(\theta_t, z^*)$$

$$\approx \eta_t \left[ \prod_{k=t+2}^{T-1} (\boldsymbol{I} - \eta_k \mathbf{H}_k) \right] \left( \boldsymbol{I} - \eta_{t+1} \sum_{z \in \mathcal{B}_{t+1}} \nabla\ell(\theta_{t+1}, z)\ell(\theta_{t+1}, z)^\top \right) \nabla\ell(\theta_t, z^*)$$

$$= \eta_t \left[ \prod_{k=t+2}^{T-1} (\boldsymbol{I} - \eta_k \mathbf{H}_k) \right] \nabla\ell(\theta_t, z^*) - \eta_t \sum_{z \in \mathcal{B}_{t+1}} \left( \eta_{t+1} \left[ \prod_{k=t+2}^{T-1} (\boldsymbol{I} - \eta_k \mathbf{H}_k) \right] \nabla\ell(\theta_{t+1}, z) \right) \nabla\ell(\theta_{t+1}, z)^\top \nabla\ell(\theta_t, z^*)$$

$$= \eta_t \left[ \prod_{k=t+2}^{T-1} (\boldsymbol{I} - \eta_k \mathbf{H}_k) \right] \nabla\ell(\theta_t, z^*) - \eta_t \sum_{z \in \mathcal{B}_{t+1}} \left( \nabla\ell(\theta_{t+1}, z)^\top \nabla\ell(\theta_t, z^*) \right) \text{DVEmb}^{(t+1)}(z)$$

$$= \eta_t \nabla\ell(\theta_t, z^*) - \eta_t \sum_{k=t+1}^{T-1} \left( \sum_{z \in \mathcal{B}_k} \left( \nabla\ell(\theta_k, z)^\top \nabla\ell(\theta_t, z^*) \right) \text{DVEmb}^{(k)}(z) \right)$$

The transition from the penultimate to the final line involves generalizing the summation over $\mathcal{B}_{t+1}$ to include all batches from $t+1$ to $T-1$, effectively unrolling the recursive computation. In other words, the "data value embedding" for data points in $t$th iteration can be approximated by its gradient subtracted by a linear combination of the data value embedding in the later iterations, where the weight of each embedding is determined by the gradient similarity $\nabla\ell(\theta_k, z)^\top \nabla\ell(\theta_t, z^*)$. $\square$

## C.4 GENERALIZED GAUSS-NEWTON APPROXIMATION TO HESSIAN

In this section, we justify the use of Generalized Gauss-Newton (GGN) as the approximation to the Hessian matrix. Similar derivation can be found in many literature and textbooks, such as Bartlett (1953); Schraudolph (2002). This approach has also been used in other data attribution and optimization techniques for approximating Hessian matrices (Martens, 2020; Kwon et al., 2023; Grosse et al., 2023).

The cross-entropy loss function for classification with one-hot encoded labels is defined as:

$$L(\mathbf{y}, \mathbf{f}) = -\sum_{i=1}^{C} y_i \log(f_i)$$

where $\mathbf{y} = [y_1, y_2, \ldots, y_C]^\top$ is the one-hot encoded true label vector and $\mathbf{f} = [f_1, f_2, \ldots, f_C]^\top$ is the vector of predicted probabilities from the model. In this paper, we restrict our focus to the cross-entropy loss, as it is the most commonly used loss function, and many LLMs are pre-trained with the cross-entropy loss function.

By chain rule, the derivative of $L$ with respect to $f_i$ is $\frac{\partial L}{\partial f_i} = -\frac{y_i}{f_i}$. Since $\mathbf{y}$ is a one-hot vector, only the correct class $k$ has $y_k = 1$, while all other $y_i = 0$ for $i \neq k$. This simplifies the gradient to:

$$\nabla_\theta L = -\frac{1}{f_k} \frac{\partial f_k}{\partial \theta}$$

Thus, the gradient depends only on the derivative of $f_k$ (the predicted probability for the correct class) with respect to $\theta$. The Hessian $\mathbf{H}$ is the second derivative of the loss with respect to $\theta$:

$$\mathbf{H} = \nabla_\theta^2 L = \frac{\partial}{\partial \theta}(\nabla_\theta L) = \frac{\partial}{\partial \theta}\left(-\frac{1}{f_k}\frac{\partial f_k}{\partial \theta}\right) = \frac{1}{f_k^2}\frac{\partial f_k}{\partial \theta}\left(\frac{\partial f_k}{\partial \theta}\right)^\top - \frac{1}{f_k}\frac{\partial^2 f_k}{\partial \theta^2}$$

Applying the product rule and assuming the second derivative $\frac{\partial^2 f_k}{\partial \theta^2}$ is negligible (which is common when $f_k$ is approximately linear in $\theta$ near the current parameter values), the Hessian simplifies to:

$$\mathbf{H} \approx \frac{1}{f_k^2}\frac{\partial f_k}{\partial \theta}\left(\frac{\partial f_k}{\partial \theta}\right)^\top$$

Moreover, this approximation matches the outer product of the gradient of loss with respect to model parameter $\theta$:

$$\nabla_\theta L \nabla_\theta L^\top = \frac{1}{f_k^2}\frac{\partial f_k}{\partial \theta}\left(\frac{\partial f_k}{\partial \theta}\right)^\top$$

Therefore, the gradient outer product exactly approximates the Hessian matrix under the assumption that $\frac{\partial^2 f_k}{\partial \theta^2}$ is negligible.

**Remark 1.** *We note that the specific definition of GGN depends on how we split the loss function. As highlighted by Kunstner et al. (2019), the GGN is ambiguous as it crucially depends on the "split" given by the inner and outer functions. For the cross-entropy loss, the canonical split treats the loss as $a_n(b) = -\log p(y_n|b)$ and $b_n(\theta) = f(x_n, \theta)$, which allows the GGN to capture curvature information to some extent. However, the alternative split used in our paper may not retain useful second-order information. Nevertheless, most recent works in influence function approximation have used this approach for approximating the Hessian thus far (with the exception of Grosse et al. (2023)). An important open question is understanding the gap between this approximation and Hessian regarding the task of data attribution.*

## C.5 GRADIENT DECOMPOSITION TECHNIQUE

To mitigate the computational cost from per-sample gradient computation, we leverage a gradient decomposition and take advantage of the computations already performed during backpropagation (Wang et al., 2025; Choe et al., 2024). We illustrate this technique with a simple linear layer, where the output is $\mathbf{s} = \mathbf{aW}$, with $\mathbf{W} \in \mathbb{R}^{d_1 \times d_2}$ being the weight matrix, $\mathbf{a} = (\mathbf{a}^{(1)}, \ldots, \mathbf{a}^{(B)})^\top$ as the input, and $\mathbf{s} = (\mathbf{s}^{(1)}, \ldots, \mathbf{s}^{(B)})^\top$ representing the pre-activation tensor. For non-sequential data,

$\mathbf{a} \in \mathbb{R}^{B \times d_1}, \mathbf{s} \in \mathbb{R}^{B \times d_2}$. Denote a sample batch as $\mathcal{B} = \{z_1, \ldots, z_B\}$. By chain rule, we can express the gradient of an individual loss $\ell^{(i)} := \ell(w, z_i)$ with respect to $\mathbf{W}$ as

$$\frac{\partial \ell^{(i)}}{\partial \mathbf{W}} = \frac{\partial \ell^{(i)}}{\partial \mathbf{s}^{(i)}} \otimes \frac{\partial \mathbf{s}^{(i)}}{\partial \mathbf{W}} = \frac{\partial \ell^{(i)}}{\partial \mathbf{s}^{(i)}} \otimes \mathbf{a}^{(i)} = \frac{\partial \ell}{\partial \mathbf{s}^{(i)}} \otimes \mathbf{a}^{(i)} \tag{12}$$

where $\ell := \sum_{j=1}^{B} \ell^{(j)}$ is the aggregated loss, and the last step is because other data points' losses have no dependency on $\mathbf{s}_i$. Note that the individual's output gradient $\frac{\partial \ell^{(i)}}{\partial \mathbf{s}^{(i)}} = \frac{\partial \ell}{\partial \mathbf{s}^{(i)}}$ is readily available during the backpropagation pass in terms of $\ell$. Therefore, this method requires only a single backpropagation on $\ell$, maintaining the training speed equivalent to standard training.

In terms of storage improvement, rather than storing the full gradient vectors $\frac{\partial \ell^{(i)}}{\partial \mathbf{W}} \in \mathbb{R}^{d_1 \times d_2}$ for each data point $z_i$, we instead store the smaller pair $\left(\mathbf{a}^{(i)}, \frac{\partial \ell}{\partial \mathbf{s}^{(i)}}\right) \in \mathbb{R}^{d_1 + d_2}$. This reduces memory requirements from $\mathcal{O}(pTB)$ to $\mathcal{O}(\sqrt{p}TB)$ for non-sequential data. For sequential data where $\mathbf{a} \in \mathbb{R}^{B \times S \times d_1}, \mathbf{s} \in \mathbb{R}^{B \times S \times d_2}$, if $S^2 > d_1 d_2$, it is more memory-efficient to directly store the per-sample gradient vectors, so the storage requirement remains as $\mathcal{O}(pTB)$.

## C.6 RANDOM PROJECTIONS FOR LARGE MODELS

For large-scale foundation models with billions of parameters, even the reduced storage of $\mathcal{O}(\sqrt{p}TB)$ can be substantial. In such cases, we apply random projections to compress the stored information further. We use two projection matrices, $\mathbf{P_a} \in \mathbb{R}^{r \times d_1}$ and $\mathbf{P_s} \in \mathbb{R}^{r \times d_2}$, to project $\mathbf{a}$ and $\frac{\partial \ell}{\partial \mathbf{s}}$ to lower dimensional space $\mathbb{R}^r$ respectively. The projected gradient can then be reconstructed directly from the projected activations and output derivatives: $(\mathbf{P_a} \otimes \mathbf{P_s})\left(\mathbf{a} \otimes \frac{\partial \ell}{\partial \mathbf{s}}\right) = (\mathbf{P_a a}) \otimes \left(\mathbf{P_s} \frac{\partial \ell}{\partial \mathbf{s}}\right)$.

## C.7 PARALLELIZED EXTENSION FOR INFLUENCE EMBEDDING COMPUTATION

The backpropagation algorithm introduced in Section 4.3 for computing data value embeddings operates with a runtime complexity of $\mathcal{O}(T)$, as it sequentially computes $\texttt{DVEmb}^{(t_s)}$ for $t_s = T-1, \ldots, 0$. While being significantly more efficient than the influence function, which requires re-computing all training gradients on the final model (see Section 5.2 and Table 2), it can still be costly for long training periods. Here, we present *influence checkpointing* technique, a parallelized extension for Algorithm 1. This extension reduces computational cost by enabling concurrent computation of embeddings at multiple checkpoints throughout the training process. Besides the computational efficiency benefits, it also enables the study of how the influence of individual data points evolves throughout model training, providing valuable insights into the learning process.

**Influence Checkpointing.** We pick $K$ evenly spaced training steps $0 < t_1 < t_2 < \ldots < t_K = T$. We then concurrently execute the backpropagation algorithm for value embedding, initiating from each of these intermediate steps. This process yields data value embeddings for each corresponding intermediate checkpoint $\theta_{t_1}, \ldots, \theta_{t_{K-1}}, \theta_{t_K}$. We extend our notation for data value embedding and denote $\texttt{DVEmb}^{(t_s \to t_\ell)}(z^*)$ as the data value embedding of $z^* \in \mathcal{B}_{t_s}$ for the intermediate checkpoint $\theta_{t_\ell}$. Note that $\texttt{DVEmb}^{(t_s)} = \texttt{DVEmb}^{(t_s \to T)}$ for the final model, and we must have $t_s < t_\ell$, as later training batches cannot influence earlier checkpoints.

Consider initiating the backpropagation algorithm in Section 4.3 at step $t_\ell$ and stop at step $t_{\ell-1}$, we will obtain data value embeddings $\texttt{DVEmb}^{(t_s \to t_\ell)}$ for $t_s = t_{\ell-1}, \ldots, t_\ell - 1$. We additionally denote $\mathbf{K}^{(t_a \to t_b)} := \prod_{t=t_a}^{t_b-1}(\mathbf{I} - \eta_t \mathbf{H}_t)$. From the definition on Equation (2), the final data value embedding $\texttt{DVEmb}^{(t_s \to T)}(z^*)$ can be computed from $\texttt{DVEmb}^{(t_s \to t_\ell)}(z^*)$ as follows:

$$\texttt{DVEmb}^{(t_s \to T)}(z^*) = \texttt{DVEmb}^{(t_s \to t_\ell)}(z^*)^\top \mathbf{K}^{(t_\ell \to T)} \tag{13}$$

Hence, to recovery of $\texttt{DVEmb}^{(t_s \to T)}(z^*)$, we additionally store the matrix $\mathbf{K}^{(t_{\ell-1} \to t_\ell)}$ between steps $t_{\ell-1}$ and $t_\ell$ during the backpropagation for each $t_\ell$. Consequently, for any $t_s$ such that $t_{\ell_0} \leq t_s < t_{\ell_0+1}$, we have $\mathbf{K}^{(t_\ell \to T)} = \prod_{\ell=\ell_0+1}^{K} \mathbf{K}^{(t_{\ell-1} \to t_\ell)}$, allowing us to compute $\texttt{DVEmb}^{(t_s \to T)}(z^*)$ based on (13). A detailed algorithm pseudocode is provided in Algorithm 2. The complexity analysis of this algorithm is the same as the original data value embedding algorithm in Table 2, but the actual runtime is being reduced by a factor of $K$ due to parallelism.

**Data Value Dynamics During Training.** In addition to its computational benefits, the influence checkpointing algorithm enables a novel capability: tracking the evolution of data influences throughout the entire model training process. If the intermediate checkpoints $\theta_{t_1}, \ldots, \theta_{t_{K-1}}$ was saved—a common practice in foundation model pretraining—we can analyze how the influence of individual data points changes as training progresses. Specifically, for any training step $t_s$ where $t_{\ell_0} \leq t_s < t_{\ell_0+1}$, we can compute the data value embedding $\mathrm{DVEmb}^{(t_s \to t_\kappa)}(z^*)$ for any checkpoint $\kappa \geq \ell_0 + 1$ as $\mathrm{DVEmb}^{(t_s \to t_\kappa)}(z^*) = \mathrm{DVEmb}^{(t_s \to t_\ell)}(z^*)^\top \left( \prod_{\ell=\ell_0+1}^{\kappa} \mathbf{K}^{(t_{\ell-1} \to t_\ell)} \right)$. This formulation allows us to estimate data influence scores not only for the final model checkpoint $\theta_T$ but for any intermediate checkpoints $\theta_{t_\kappa}$. As a result, we gain a more fine-grained and dynamic view of how data influences evolve during training, providing deeper insights into the model's learning behavior over time. To our knowledge, this is the first data attribution method to offer such a principled and practical framework for studying data influence dynamics throughout the training process. This capability opens up new avenues for understanding and optimizing machine learning model training.

## C.8 PSEUDOCODE

---
**Algorithm 1** Backpropagation for computing data value embedding from the final checkpoint

---
**Require:** Training steps $T$, learning rates $\{\eta_t\}_{t=0}^{T-1}$, training data gradients $\{\nabla\ell(\theta_t, z)\}_{t=0, z \in \mathcal{B}_t}^{T-1}$
1: // Initialization
2: $\mathbf{M}^{(T-1)} \leftarrow \mathbf{0}$.
3:
4: // Recursion steps
5: **for** $t = T - 1$ **down to** $0$ **do**
6:     **for** $z \in \mathcal{B}_t$ **do**
7:         $\mathrm{DVEmb}^{(t)}(z) \leftarrow \eta_t \nabla\ell(\theta_t, z) - \eta_t \mathbf{M}^{(t)} \nabla\ell(\theta_t, z)$
8:     **if** $t > 0$ **then**
9:         $\mathbf{M}^{(t-1)} \leftarrow \mathbf{M}^{(t)} + \sum_{z \in \mathcal{B}_t} \mathrm{DVEmb}^{(t)}(z) \nabla\ell(\theta_t, z)^\top$
10: **return** $\{\mathrm{DVEmb}^{(t)}(z)\}_{t=0, z \in \mathcal{B}_t}^{T-1}$

---

---

**Algorithm 2** Parallel Influence Checkpointing for Data Value Embedding

---

**Require:** Training steps $T$, number of checkpoints $K$, learning rates $\{\eta_t\}_{t=0}^{T-1}$, loss gradients $\{\nabla\ell(\theta_t, z)\}_{t=0, z\in\mathcal{B}_t}^{T-1}$, Hessians $\{\mathbf{H}_t\}_{t=0}^{T-1}$
**Ensure:** Data value embeddings $\{\texttt{DVEmb}^{(t)}(z)\}_{t=0, z\in\mathcal{B}_t}^{T-1}$
1: Select $K$ evenly spaced checkpoints $0 = t_0 < t_1 < t_2 < \ldots < t_K = T$
2: **for** $\ell = 1$ **to** $K$ **do**
3:     Run BACKPROPAGATESEGMENT$(t_{\ell-1}, t_\ell)$

4:
5: // Compute final embeddings
6: **for** $\ell = 1$ **to** $K$ **do**
7:     **for** $t_s = t_{\ell-1}$ **to** $t_\ell - 1$ **do**
8:         **for** $z \in \mathcal{B}_{t_s}$ **do**
9:             $\texttt{DVEmb}^{(t_s)}(z) \leftarrow \texttt{DVEmb}^{(t_s \to t_\ell)}(z)^\top \prod_{k=\ell+1}^{K} \mathbf{K}^{(t_{k-1} \to t_k)}$
10: **return** $\{\texttt{DVEmb}^{(t)}(z)\}_{t=0, z\in\mathcal{B}_t}^{T-1}$
11:
12: **procedure** BACKPROPAGATESEGMENT$(t_a, t_b)$
13:     Initialize and $\mathbf{M}^{(t_b-1)}$ as in the original algorithm
14:     $\mathbf{K}^{(t_b \to t_b)} \leftarrow \boldsymbol{I}$
15:     **for** $t = t_b - 1$ **down to** $t_a$ **do**
16:         **for** $z \in \mathcal{B}_t$ **do**
17:             $\texttt{DVEmb}^{(t \to t_b)}(z) \leftarrow \eta_t \nabla\ell(\theta_t, z) - \eta_t \mathbf{M}^{(t)} \nabla\ell(\theta_t, z)$
18:         **if** $t > t_a$ **then**
19:             $\mathbf{M}^{(t-1)} \leftarrow \mathbf{M}^{(t)} + \sum_{z\in\mathcal{B}_t} \texttt{DVEmb}^{(t \to t_b)}(z) \nabla\ell(\theta_t, z)^\top$
20:         $\mathbf{K}^{(t \to t_b)} \leftarrow \mathbf{K}^{(t+1 \to t_b)}(\boldsymbol{I} - \eta_t \mathbf{H}_t)$
21:     **return** $\{\texttt{DVEmb}^{(t \to t_b)}(z)\}_{t=t_a, z\in\mathcal{B}_t}^{t_b-1}, \mathbf{K}^{(t_a \to t_b)}$

---

## C.9 COMPLEXITY SUMMARY

In this section, we compare the storage, memory, and computational efficiency of data value embedding with LoGRA (Choe et al., 2024), the most efficient implementation of the influence function to date. LoGRA is currently the only method that supports real-time, on-demand data influence computation when new test data is introduced. Similar to our algorithm, LoGRA initially computes per-sample training gradients on the final model for *all* training data points $z^* \in \mathcal{D}$, where $\mathcal{D}$ represents the dataset. It then stores the *projected* Hessian-adjusted gradients $\mathbf{H}_T^{-1}\nabla\ell(\theta_T, z^*)$ for each $z^*$, and also assumes layer gradient independence.

While the random projection step in LoGRA is akin to our approach, LoGRA's requirement to recompute gradients for all training data on the final model $\theta_T$ is computationally intensive, effectively equivalent to one epoch of model training. In contrast, data value embedding captures the training data gradients during the original training process. As discussed in Appendix C.10, the training and disk storage can be handled asynchronously. This means that the gradient storage step in the data value embedding algorithm does not incur additional efficiency costs.

| | Storing $\mathbf{H}_T^{-1}\nabla\ell(\theta_T, z^*)$ / data value embedding | | | Compute Influence (dot-product) | |
| --- | --- | --- | --- | --- | --- |
| | Storage | Memory | FLOPS | Memory | FLOPS |
| LoGRA | $\mathcal{O}(\|\mathcal{D}\|\tilde{p})$ | $\mathcal{O}(p)$ | $\|\mathcal{D}\|p + \|\mathcal{D}\|\sqrt{p\tilde{p}}/L$ | $\mathcal{O}((B_{\text{test}} + B_{\text{train}})\tilde{p})$ | $\mathcal{O}(B_{\text{test}}B_{\text{train}}\tilde{p})$ |
| Data Value Embedding | $\mathcal{O}(TB\tilde{p})$ | $\mathcal{O}(p)/\mathcal{O}(B\tilde{p}^2/L^2)$* | $TB\sqrt{p\tilde{p}}/L/\mathcal{O}(TB\tilde{p}^2/L)$* | $\mathcal{O}((B_{\text{test}} + B_{\text{train}})\tilde{p})$ | $\mathcal{O}(B_{\text{test}}B_{\text{train}}\tilde{p})$ |

Table 2: Summary of the storage, memory, and FLOPS complexity for LoGRA (Choe et al., 2024), the most efficient implementation of the influence function to date. Here, $p$ denotes the model dimension, $\tilde{p}$ is the projected dimension, $T$ represents the number of training iterations, and $B$ is the batch size. $|\mathcal{D}|$ is the dataset size, with the relationship $TB = |\mathcal{D}| \times \text{#epochs}$. $L$ is the number of layers. $B_{test}$ and $B_{train}$ refer to the test and training batch sizes during influence computation, respectively, which are independent of the batch size $B$ used during model training. *Since the data value embedding technique involves two distinct steps for storing relevant information for data attribution (storing gradients during training & computing/storing data value embeddings after training), we include the complexity for both steps. For the gradient storage step, the complexity refers to the *additional* cost beyond regular training.

## C.10 PRACTICAL CONSIDERATIONS & POTENTIAL EXTENSIONS

In this section, we discuss some practical extensions and considerations for implementing data value embedding for real-world scenarios.

**Optimizing I/O operations for seamless training.** During each training iteration, computed gradient representations need to be transferred from GPU to CPU and then written to disk. To prevent this process from blocking the main training loop, we implement several optimizations: **(1) Asynchronous I/O operations**: To avoid the gradient storing process blocking the main training loop, we make GPU operations and GPU-CPU transfers asynchronous by using CUDA streams. This allows GPU computations to continue while data is being transferred. We also offload the disk write process to a separate thread or process, allowing the main training loop to proceed without waiting for disk operations to complete. **(2) Gradient accumulation**: Instead of writing gradients to disk after every iteration, we can accumulate them over multiple iterations and then write them in bulk. This reduces the frequency of disk I/O operations, improving overall efficiency.

**Approximating data value embeddings from checkpoints alone.** In situations where only intermediate model checkpoints are accessible and per-training-step (projected) gradient vectors are unavailable—such as when modifying the training loop's implementation is impossible or when disk storage is limited—we can adapt our approach by assuming that there is only one gradient update step between each checkpoint, similar to assumptions made in other data attribution literature (Pruthi et al., 2020). Under this assumption, we compute the gradient for each training point at the checkpoint immediately following its corresponding training iteration. These estimated gradients are then used to execute the backpropagation algorithm, enabling the computation of data value embeddings without requiring gradient saving during the original training run.

**Dataset-level attribution through embedding aggregation.** In practical applications, stakeholders often require valuation at the dataset level rather than for individual data points. To address this need,

a natural extension of our approach is to compute the data value embedding for a dataset by summing the data value embeddings of all constituent data points from the same source. This method offers a significant advantage over the summation of expected LOO scores, as it inherently accounts for complex inter-data interactions throughout the training process. However, data value embeddings are derived based on first-order Taylor approximations. While these approximations are accurate for estimating small perturbations to the model, making them suitable for predicting the effects of removing individual training points, their accuracy may diminish when aggregating over larger sets of data points. The potential discrepancy between individual-level accuracy and dataset-level aggregation presents an interesting avenue for future research.

# D INFLUENCE FUNCTION AS AN INADEQUATE APPROXIMATION FOR THE EXPECTED LEAVE-ONE-OUT SCORE

**Expected LOO.** The expected LOO is an alternative to traditional LOO that has been discussed in the past literature (Feldman & Zhang, 2020). The *expected LOO* is the trajectory-specific LOO averaged over all possible training runs characterized by different random initializations and mini-batch selections. Formally, it is defined as $\texttt{ELOO}(z^*; z^{(val)}) := \mathbb{E}_\omega \left[ \ell(\theta'_T(\omega), z^{(val)}) - \ell(\theta_T(\omega), z^{(val)}) \right]$ where $\theta_T(\omega)$ and $\theta'_T(\omega)$ denote the final model parameters obtained with and without $z^*$, respectively, under the randomness $\omega$ which encodes the choices of training batch order and parameter initialization. While the expected LOO offers a general assessment of a data point's influence by averaging over multiple training runs, it may obscure the variability introduced by stochastic training dynamics. In contrast, by accounting for factors such as random initialization and mini-batch selection in a specific run, we argue that the trajectory-specific LOO provides a fine-grained assessment of a data point's impact *for the trained model*. This is particularly important in practical scenarios, such as deploying a specific model for production, where stakeholders are interested in the valuation of data with respect to that specific deployed model rather than the general learning algorithm.

While the influence function provides valuable insights, it overlooks the specific training trajectory. This raises the question: *Can the influence function be interpreted as an estimate of the trajectory-specific leave-one-out score?*

We consider the following training batch sampling process. Let $\sigma := (\mathcal{B}_0, \dots, \mathcal{B}_{T-1})$ represent a fixed sequence of training batches formed from the leave-one-out dataset $\mathcal{D} \setminus z^*$. The training point $z^*$ is uniformly likely to be added to any one of the training batches in $\sigma$. Additionally, denote $\sigma^{(t_s)} := (\mathcal{B}_0, \dots, \mathcal{B}_{t_s} \cup \{z^*\}, \dots, \mathcal{B}_{T-1})$ as the training batch sequence where $z^*$ is incorporated into the $t_s$-th batch. Let $\theta_k^{(\sigma^{(t_s)})}$ denote the model parameters at the $k$-th iteration when training with batch sequence $\sigma^{(t_s)}$, and let $\mathbf{H}_k^{(\sigma^{(t_s)})}$ denote the Hessian matrix at the $k$-th iteration when training on sequence $\sigma^{(t_s)}$.

When the specific training iteration $t_s$ where the training point of interest $z^*$ is added is unknown, it is natural to estimate the expected influence score across all possible scenarios. The expected influence score for $z^*$ based on the unrolling differentiation approximation is given by:

$$\mathbb{E}_{t_s \sim [T-1]} \left[ \eta_{t_s} \nabla \ell(\theta_T^{(\sigma^{(t_s)})}, z^{(val)})^\top \left[ \prod_{k=t_s+1}^{T-1} (\boldsymbol{I} - \eta_k \mathbf{H}_k^{(\sigma^{(t_s)})}) \right] \nabla \ell(\theta_{t_s}^{(\sigma^{(t_s)})}, z^*) \right] \qquad (14)$$

**Theorem 5** (Influence Function Approximation). *Under the following assumptions: **(1) Model Approximation:** $\theta_k^{(\sigma^{(t_s)})} \approx \theta_T$ for all $t_s$ and $k = 0, \dots, T-1$; **(2) Hessian Approximation:** $\mathbf{H}_k^{(\sigma^{(t_s)})} \approx \mathbf{H}_T^{(*)} := \frac{1}{T} \sum_{z \in \mathcal{D}} \nabla^2 \ell(\theta_T, z)$ for all $k$; **(3) Constant Learning Rate:** $\eta_t = \eta$ for all $t = 0, \dots, T-1$; the expected influence score in Equation* (14) *simplifies and converges to the standard influence function formulation for large* $T$:*

$$(14) \approx \nabla \ell(\theta_T, z^{(val)})^\top \left( \sum_{z \in \mathcal{D}} \nabla^2 \ell(\theta_T, z) \right)^{-1} \nabla \ell(\theta_T, z^*)$$

**Implications.** The derivation demonstrates that, under the stated approximations, influence function effectively approximates the expected influence score derived from the unrolling differentiation approach as $T$ becomes large. This approximation indicates that the influence function may not fully represent the true leave-one-out score because it relies on simplifying assumptions—such as approximating all model checkpoints and Hessian matrices to be identical—that often do not hold in practical training scenarios. In real-world settings, model parameters evolve significantly throughout training, learning rates are typically scheduled to change over time, and the Hessian matrices can vary considerably between iterations. These factors undermine the validity of the assumptions, thereby limiting the effectiveness of the influence function as an approximation for the leave-one-out score.

*Proof.* Assume that all we have access to is the final model checkpoint $\theta_T := \theta_T^{(\sigma^{(t_r)})}$ for a specific realization where $t_s = t_r \sim [T-1]$. Under this assumption, the best approximation we can make is:

$$\theta_T^{(\sigma^{(t_s)})} \approx \theta_k^{(\sigma^{(t_s)})} \approx \theta_T$$

for any $t_s$ and $k = 0, \ldots, T-1$. Additionally, we approximate the Hessian matrices as:

$$\mathbf{H}_k^{(\sigma^{(t_s)})} \approx \mathbf{H}_T^{(*)} := \frac{1}{T} \sum_{z \in \mathcal{D}} \nabla^2 \ell(\theta_T, z) \tag{15}$$

and assume a constant learning rate $\eta_t = \eta$ for all $t = 0, \ldots, T-1$.

With these approximations, Equation (14) simplifies to:

$$(14) = \mathbb{E}_{t_s \sim [T-1]} \left[ \eta \nabla \ell(\theta_T, z^{(val)})^\top \left[ \prod_{k=t_s+1}^{T-1} (\mathbf{I} - \eta \mathbf{H}_T^{(*)}) \right] \nabla \ell(\theta_T, z^*) \right] \tag{16}$$

$$= \eta \nabla \ell(\theta_T, z^{(val)})^\top \mathbb{E}_{t_s \sim [T-1]} \left[ (\mathbf{I} - \eta \mathbf{H}_T^{(*)})^{T-1-t_s} \right] \nabla \ell(\theta_T, z^*) \tag{17}$$

$$= \frac{\eta}{T} \nabla \ell(\theta_T, z^{(val)})^\top \sum_{t_s=0}^{T-1} \left( (\mathbf{I} - \eta \mathbf{H}_T^{(*)})^{T-1-t_s} \right) \nabla \ell(\theta_T, z^*) \tag{18}$$

$$\approx \frac{\eta}{T} \nabla \ell(\theta_T, z^{(val)})^\top \left( \eta \mathbf{H}_T^{(*)} \right)^{-1} \nabla \ell(\theta_T, z^*) \tag{19}$$

$$= \nabla \ell(\theta_T, z^{(val)})^\top \left( \sum_{z \in \mathcal{D}} \nabla^2 \ell(\theta_T, z) \right)^{-1} \nabla \ell(\theta_T, z^*) \tag{20}$$

The approximation in the fourth step arises from summing the geometric series of matrices. For $\eta$ sufficiently small and $T$ large, we have $\sum_{s=0}^{T-1} (\mathbf{I} - \eta \mathbf{H}_T^{(*)})^s \approx (\eta \mathbf{H}_T^{(*)})^{-1}$. $\qquad \square$

# E  ADDITIONAL EXPERIMENTS

## E.1  BASELINE & IMPLEMENTATION DETAILS

**Fidality Evaluation (Section 5.1).** Given the computational intensity, we conduct our experiments on a subset (10%) of the MNIST using an MLP with two layers with 128 neurons in the hidden layer. We train the model with standard SGD with a learning rate $10^{-2}$ for 10 epochs. We randomly pick 100 data points and compute their ground-truth trajectory-specific LOO score. For the single epoch removal, we remove the data point from the last epoch. For this experiment, we do not use random projection and use the full gradients. For the comparison with influence function, we use the state-of-the-art implementation from LoGRA (Choe et al., 2024) with the damping term set to be $10^{-3}$ following Bae et al. (2022).

**Large-scale Experiments in Section 5.2, 5.3 and E.4.** Our experiments focus on two language models: Pythia-410M and GPT2-Small. We train these models on two commonly used datasets in the literature for large-scale language model training: **(1)** A 1% subset of the Pile dataset (Gao et al., 2020), and **(2)** Wikitext-103 (Merity et al., 2016). We note that our choice of model architecture size is primarily constrained by the available GPUs in our current setup. However, this limitation does not diminish the significance of our findings. With enough computational resources (e.g., 8 H100 GPUs), our method is readily applicable to perform data attribution for billion-scale model training.

For both settings, the sequence length is set to 1024. The learning rate is set at a maximum of $3 \times 10^{-4}$. We use AdamW as the optimizer with a weight decay of 0.1, and beta values set to 0.9 and 0.95. Gradients are clipped at a maximum value of 1.0 to maintain stability during training. The batch size is set to 16, with a learning rate warmup of 2000 iterations followed by cosine decay.

For all experiments, for storage reasons, we compute and store projected gradients and data value embedding on linear layers of the model only, with the projection dimension set to 1024 per layer. However, we stress that this is not restricted by computation but by disk storage limit.

## E.2  ADDITIONAL RESULTS FOR FIDELITY EVALUATION (SECTION 5.1)

**Evaluation on more epochs.** Here, we show additional results for the fidelity experiment in Section 5.1, where the model is being trained for a longer time (10 epochs), in which case the model is closer to convergence. Figure 5 shows that even in this case, data value embedding still has a high Spearman correlation with the ground-truth LOO in both settings, and the influence function exhibits almost no correlation with the LOO score.

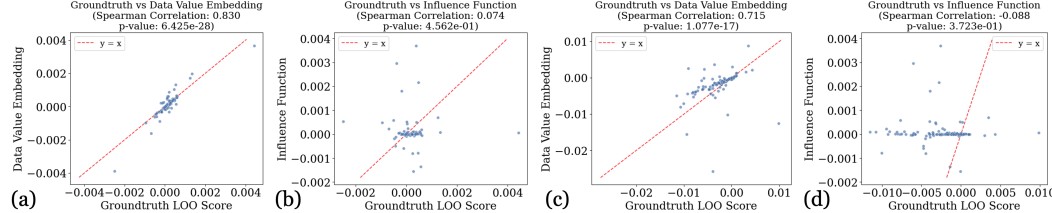

Figure 5: The correlation between ground-truth LOO when the MLP is trained for **10 epochs** and the estimation obtained by (a) the data value embedding method and (b) the influence function for *single epoch removal*. (c) and (d) present the corresponding correlations for *all-epoch removal*.

**Evaluation on different architectures.** To demonstrate that our method's effectiveness extends beyond simple MLPs, we evaluate data value embedding on a CNN architecture consisting of two convolutional layers (with 32 and 64 filters, respectively, each followed by 2x2 max pooling) and a final linear layer. We train the model on MNIST using SGD with learning rate $10^{-2}$ for 10 epochs. Following the same experimental setup as with MLP, we randomly select 100 data points and compute their ground-truth trajectory-specific LOO scores for both single-epoch and all-epochs removal settings. Figure 6 shows that data value embedding maintains strong correlation with ground-truth LOO scores, achieving Spearman correlations of 0.818 for single-epoch removal (Figure 6 (a)) and 0.682 for all-epochs removal (Figure 6 (b)). These results demonstrate that our method can effectively approximate data influence across different neural architectures.

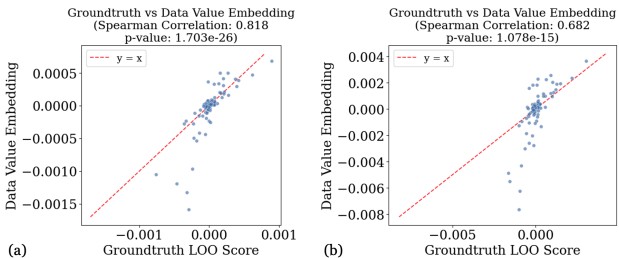

Figure 6: Scatter plot showing the correlation between ground-truth LOO and data value embedding method when training a small CNN on MNIST for **10 epochs**, where (a) is for *single epoch removal* (b) for *all-epoch removal* setting.

### E.2.1 Effectiveness on Mislabeled Data Detection and Data Selection

In addition to comparing our results with ground-truth training run-specific LOO, we further evaluate the performance of our data value embedding algorithm in the task of mislabeled data detection and data selection, two standard benchmarks in data attribution literature. We compare several data attribution baselines, including *Retraining-based Data Shapley (Ghorbani & Zou, 2019), KNN-Shapley (Jia et al., 2019), Influence Function (Koh & Liang, 2017), Trak (Park et al., 2023), Empirical Influence Functions (Feldman & Zhang, 2020), and Datamodels (Ilyas et al., 2022).*

**Experiment settings.** We use ImageNet-pretrained ResNet18 as the architecture in the experiment. Given the computational intensity of retraining-based methods, we conduct our experiments on a subset of 1,000 samples from CIFAR-10 dataset. We use Adam with a learning rate 0.001, weight decay of 1e-4, and label smoothing of 0.1 over 50 epochs. The learning rate is reduced by a factor of 0.1 every 10 epochs. The batch size is set to 64. For retraining-based techniques (Retraining-based Data Shapley, Empirical Influence Functions, Datamodels), we estimate the corresponding attribution scores with 1000 model training runs. For Trak, we set the projection dimension to be 2048. For KNN-Shapley, we set $K = 5$ and use the features extracted from the last linear layer of ResNet18. Both experiments included 10% random label noise to reflect the challenges of real-world data.

**Remark 2.** *The mislabeled data detection and data selection benchmark here mainly serves to evaluate the fidelity of our algorithm in settings where ground-truth LOO computation is infeasible. However, we stress that data value embedding is not specifically designed for those tasks. Rather, it is developed as an interpretability tool and a mechanism for real-time data valuation, with potential applications in data marketplaces and addressing AI copyright disputes (Wang et al., 2024).*

**I. Mislabeled Data Detection.**

Table 3 shows that KNN-Shapley achieves the highest accuracy in detecting mislabeled data, likely due to its sensitivity to label inconsistencies. *Retraining-based methods* (Retraining-based Data Shapley, Empirical Influence Functions, Datamodels) exhibit the lowest performance, which can be attributed to the inefficiency of Monte Carlo sampling and the inherent stochasticity during retraining, as discussed in Wang & Jia (2023a). Among techniques requiring only a single training run, Trak underperforms relative to other methods. This observation aligns with findings from its original paper (Park et al., 2023), which suggests that ensemble methods are often necessary for optimal performance. Notably, data value embedding and influence function achieve comparable performance, outperforming all other techniques except KNN-Shapley. The strong performance of these methods likely stems from their deterministic nature, which provides more consistent and reliable results.

**II. Data Selection.**

Table 4 demonstrates that Data Value Embedding outperforms *all* existing data valuation methods in the task of data selection. *Retraining-based methods* (Data Shapley, Empirical Influence Functions, Datamodels) show limited effectiveness due to the high variance introduced by Monte Carlo sampling and learning stochasticity. While the influence function and Trak do not require model retraining, their performance is constrained by assumptions that often do not hold in practice, such as model convergence and strong convexity. KNN-Shapley provides stable valuation results. However, it assigns similar scores to similar data points, potentially reducing dataset diversity among the selected data subset. In contrast, Data Value Embedding considers both data characteristics and temporal

| Method | Performance (Mean ± Std) |
|---|---|
| **Data Shapley** (Ghorbani & Zou, 2019) | 0.582 (0.029) |
| **Empirical Influence Function** (Feldman & Zhang, 2020) | 0.552 (0.017) |
| **Datamodels** (Ilyas et al., 2022) | 0.520 (0.008) |
| **KNN-Shapley** (Jia et al., 2019; Wang & Jia, 2023b) | 0.760 (0.018) |
| **Trak** (Park et al., 2023) | 0.511 (0.012) |
| **Influence Function** (Koh & Liang, 2017) | 0.654 (0.054) |
| **Data Value Embedding (ours)** | 0.667 (0.031) |

Table 3: AUROC scores of mislabeled data detection task with various data attribution techniques on CIFAR10 dataset. The higher the AUROC score is, the better the method is. The results are across three different training runs (the randomness comes from construction of corrupted datasets), where we show the standard deviation in ().

ordering in training, allowing similar data points to receive different scores based on when they appear in the training sequence. This temporal awareness helps maintain dataset diversity while identifying valuable samples.

| | 20% | 40% | 60% | 80% |
|---|---|---|---|---|
| **Random** | 0.350 (0.010) | 0.461 (0.010) | 0.525 (0.004) | 0.559 (0.003) |
| **Data Shapley** (Ghorbani & Zou, 2019) | 0.317 (0.047) | 0.468 (0.010) | 0.527 (0.004) | 0.570 (0.008) |
| **Empirical Influence Function** (Feldman & Zhang, 2020) | 0.342 (0.004) | 0.466 (0.016) | 0.530 (0.009) | 0.568 (0.010) |
| **Datamodels** (Ilyas et al., 2022) | 0.342 (0.004) | 0.465 (0.004) | 0.534 (0.010) | 0.559 (0.005) |
| **KNN-Shapley** (Jia et al., 2019) | 0.354 (0.017) | 0.478 (0.007) | 0.525 (0.015) | 0.563 (0.005) |
| **Trak** (Park et al., 2023) | 0.329 (0.021) | 0.443 (0.030) | 0.517 (0.016) | 0.572 (0.009) |
| **Influence function** (Koh & Liang, 2017) | 0.320 (0.033) | 0.450 (0.028) | 0.530 (0.015) | 0.580 (0.004) |
| **Data Value Embedding (ours)** | **0.391 (0.007)** | **0.518 (0.008)** | **0.566 (0.005)** | **0.604 (0.009)** |

Table 4: Test accuracies when training ResNet18 on high-value data points selected by various data attribution techniques. To be able to compare with techniques that require model retraining, for each training run we randomly sample a size-1000 subset of CIFAR10 dataset (with 10% data points being mislabeled). The results are across three different training runs (the randomness comes from construction of corrupted datasets), where we show the standard deviation in ().

### E.3 ADDITIONAL DISCUSSION AND RESULTS FOR SECTION 5.3

#### E.3.1 EXPLANATION OF INFLUENCE TREND

**1. Parameter initialization and warmup training are important for the final model performance.** The blue curve in Figure 7 (b) illustrates the trend of average training data gradient norm throughout the training process. We observe that gradient norms are typically large and unstable during early training ($t_s \leq 2000$). As training progresses, these norms decrease rapidly, leading to a significant reduction in the eigenvalues of the Hessian matrix $\mathbf{H}_t \approx \sum_{z \in \mathcal{B}_t} \nabla \ell(\theta_t, z) \nabla \ell(\theta_t, z)^\top$. Consequently, when $\|\nabla \ell(\theta_{t_s})\|$ is significantly larger than later training gradients, the norm of data value embedding $\left\| \prod_{t=t_s+1}^{T} (\boldsymbol{I} - \eta_t \mathbf{H}_t) \nabla \ell(\theta_{t_s}) \right\|$ remains substantial. This results in early-stage data points maintaining significant influence until the end of training. Figure 7 (a) further illustrates this phenomenon. The purple curve shows that training data points from the High-impact Warmup Phase, while experiencing large drops in influence as training progresses, still maintain higher influence than later data points. This observation aligns with the well-established effect that model initialization and/or warm-up training plays a crucial role in training performance, effectively initializing model parameters and gradually preparing the model for more complex learning tasks.

**2. Influence saturation from future data.** As the model enters a relatively smooth loss regime ($t_s > 2000$ in Figure 7 (b)), the training data gradient norm decreases much more slowly. In this phase, the magnitude deflation effect from $\prod_{t=t_s}^{T} (\boldsymbol{I} - \eta_t \mathbf{H}_t)$ remains significant for relatively small $t_s$, while the training gradient norm $\|\nabla \ell(\theta_{t_s})\|$ does not differ significantly between earlier and later training points. This results in $\left\| \prod_{t=t_s}^{T} (\boldsymbol{I} - \eta_t \mathbf{H}_t) \nabla \ell(\theta_{t_s}) \right\| < \|\nabla \ell(\theta_{t_a})\|$ for $t_a > t_s$, creating a low-impact basin during the early-to-middle training stage. In this basin, influence scores are lower than those of data points from both the very early and later training stages. The red curve in Figure 7

(a) demonstrates this trend, showing influence scores for these points gradually decreasing during training and eventually falling below those of later training data points. This pattern aligns with the phenomenon of catastrophic forgetting, where the model appears to "forget" the influence of data from this middle phase as training progresses. One might initially think this phenomenon is connected to catastrophic forgetting, where the model appears to "forget" the influence of data from earlier training phases as it progresses. However, we note that a data point's influence score decreases the most when future data points are similar to it, which is different from catastrophic forgetting. Intuitively, if future points are identical, the presence of the earlier data point in training becomes less relevant to the model's behavior.

In Figure 7(b), we consider a simplified setting where we approximate Hessian with GGN matrix and assume all training gradients are orthogonal across different iterations. Under these assumptions, $\mathbf{H}_k \approx \boldsymbol{G}_k = \sum_{z \in B_k} \nabla\ell(\theta_k, z)\nabla\ell(\theta_k, z)^\top$ becomes a sum of rank-1 matrices that have non-overlapping eigenspaces. Given the orthogonality assumption, we have $\boldsymbol{G}_t\boldsymbol{G}_s = 0$ for $t \neq s$, and the product $\prod_{t=t_s}^{T}(I - \eta_t\boldsymbol{G}_t)$ simplifies to $I - \sum_{t=t_s}^{T}\eta_t\boldsymbol{G}_t$. Since each $\boldsymbol{G}_t = \sum_{z \in B_t}\nabla\ell(\theta_t, z)\nabla\ell(\theta_t, z)^\top$ is a sum of rank-1 matrices along orthogonal directions, the trace of this product can be analytically computed as $p - \sum_{t=t_s}^{T}\eta_t\sum_{z \in B_t}\|\nabla\ell(\theta_t, z)\|^2$, where $p$ is the dimension of parameter space. Furthermore, if we assume $\nabla\ell(\theta_{t_s})$ follows a Gaussian distribution, then $\left\|(I - \sum_{t=t_s}^{T}\eta_t G_t)\nabla\ell(\theta_{t_s})\right\|$ follows a scaled chi distribution since it's the norm of a Gaussian vector after linear transformation by an orthogonal projection matrix. This enables us to analytically compute its expected value, as shown by the green curve in Figure 7 (b).

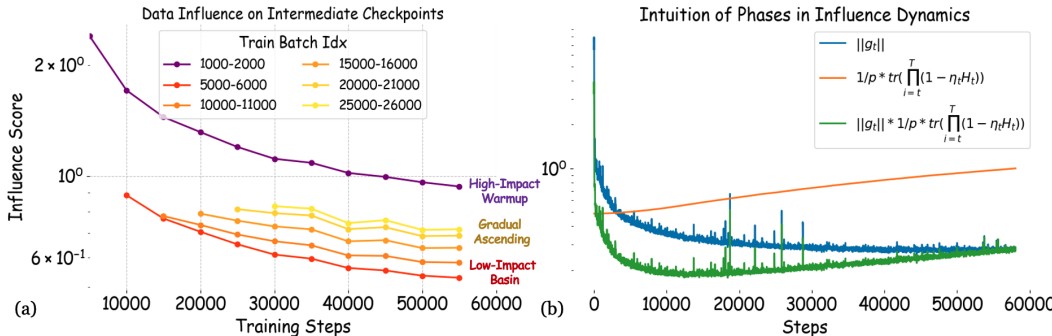

Figure 7: **(a)** (same as Figure 4 in the main paper) Influence scores of data points from different training stages on intermediate model checkpoints throughout training. The x-axis denotes the number of training iterations, and the y-axis represents the influence score of selected data points on the model at each checkpoint. **(b)** The blue curve shows the average gradient changes as model training progresses. The orange and green curves are analytical curves under a simplified setting, where the orange curve is the analytical trace of $\prod_{t=t_s}^{T}(\boldsymbol{I} - \eta_t\mathbf{H}_t)$ as $t_s$ increases, and the green curve shows the norm of data value embedding for Gaussian-distributed gradient under this simplified setting.

### E.3.2 ADDITIONAL DETAILS FOR FIGURE 1 (B)

In Figure 1(b), we compare different strategies for applying online data selection during model training. The online selection process identifies high-quality training batches by (1) sampling a candidate pool of training points with size 2B, where B is the desired batch size, (2) computing the gradient cosine similarity between each candidate point and a small validation batch (randomly sampled from the full validation set), and (3) selecting the B points with the highest similarity scores to form the next training batch. This procedure incurs significant computational overhead, requiring additional forward and backward passes for similarity computation at each selection step. When not performing online selection (i.e., during the "random selection" phases), we simply sample training batches randomly. Notably, the model processes the entire training dataset regardless of the selection strategy - what varies is only how batches are prioritized during different training phases. The "Early+Late" strategy applies online selection only during iterations 1-2000 and after iteration 20000, while using random selection in between. This selective approach achieves 96% of the performance

improvement of continuous selection while reducing the computational overhead by more than 5×, suggesting that precise batch selection is most critical during the early and late training phases.

### E.3.3 ADDITIONAL RESULTS

Figure 8 presents additional results on the data influence scores of training data across different stages of LLM pretraining, using more datasets and model architectures. We observe that the data influence scores on the final model can consistently be categorized into three distinct regimes throughout pretraining.

Figure 9 shows the results when using pretrained models downloaded from Huggingface. In this scenario, the situation diverges across different datasets. Notably, when we continually pretrain on the Pile dataset, no gradual ascending phase is observed at the end. However, when GPT-2 has already been pre-trained, continuing pretraining on Wikitext-103 once again exhibits a gradual ascending phase. This is likely because Wikitext-103 is a relatively small dataset, and fine-tuning it for three epochs can easily lead to overfitting, as illustrated in Figure 10 (d).

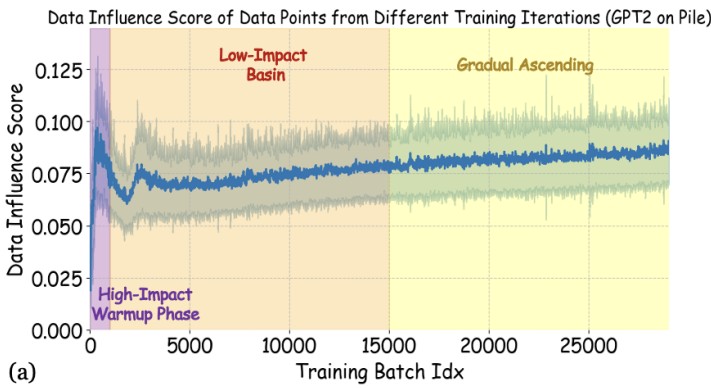

(a)

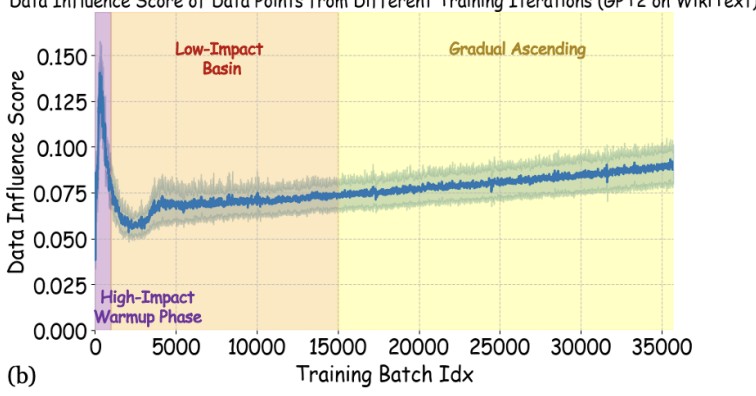

(b)

Figure 8: Average data influence scores per training batch, measured against the final model's loss where (a) GPT2 trained on 1% of Pile, and (b) GPT2 trained on WikiText-103M for 3 epochs.

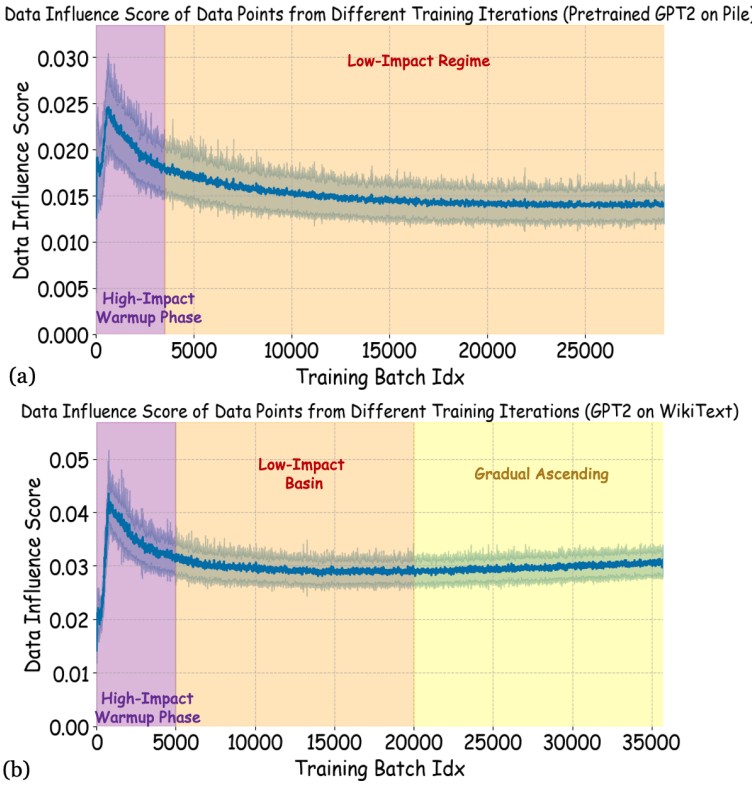

Figure 9: Average data influence scores per training batch, measured against the final model's loss where (a) Pretrained GPT2 trained on 1% of Pile, and (b) Pretrained GPT2 trained on WikiText-103M for 3 epochs.

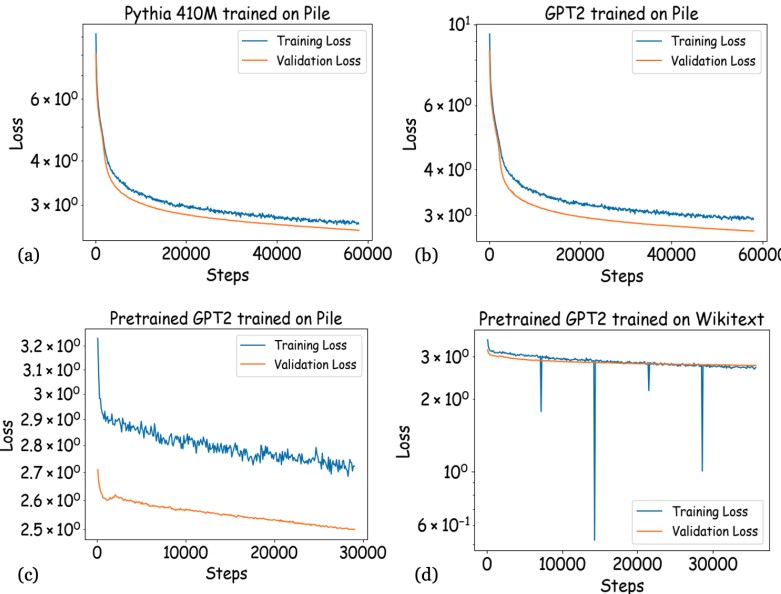

Figure 10: Loss curve for the training.

E.4 QUALITATIVE EVALUATION

We conduct a qualitative analysis to examine the similarities between a test data point $z^{(val)}$ and the most valuable data points identified by data value embedding. In this experiment, we set $z^{(val)}$ to be identical to one of the training data points, making the most similar data point its own repetition. In data valuation literature, the influence score of a training point on its repetition is usually referred to as "self-influence" (Koh & Liang, 2017) and is being used to measure memorization (Feldman & Zhang, 2020). Intuitively, the self-influence should be the highest among all training points.

Figure 11 shows representative results from training GPT-2 on Wikitext-103 over three epochs, where the test data is about  military video game . As observed, for model checkpoints after the 2nd and 3rd epochs, the  test data point's repetition  achieves the highest influence score, as expected. However, for the model checkpoint after the 1st epoch, the most valuable data points are not the repetition but rather a similar data about  war history . This discrepancy occurs because, during the first epoch of training, the repetition of the test data point resides in the *low-value basin* identified in Section 5.3, resulting in a lower self-influence score as subsequent training progresses. Additionally, we observe that the influence function may incorrectly identify irrelevant data points as highly influential (e.g., the  Popular Music  completely irrelevant to  military video game  but being identified as the second most valuable data), possibly due to its bias towards data points with high gradient norms, as also noted in Barshan et al. (2020). This limitation underscores the advantages of data value embedding in providing more accurate and context-aware data influence assessment.

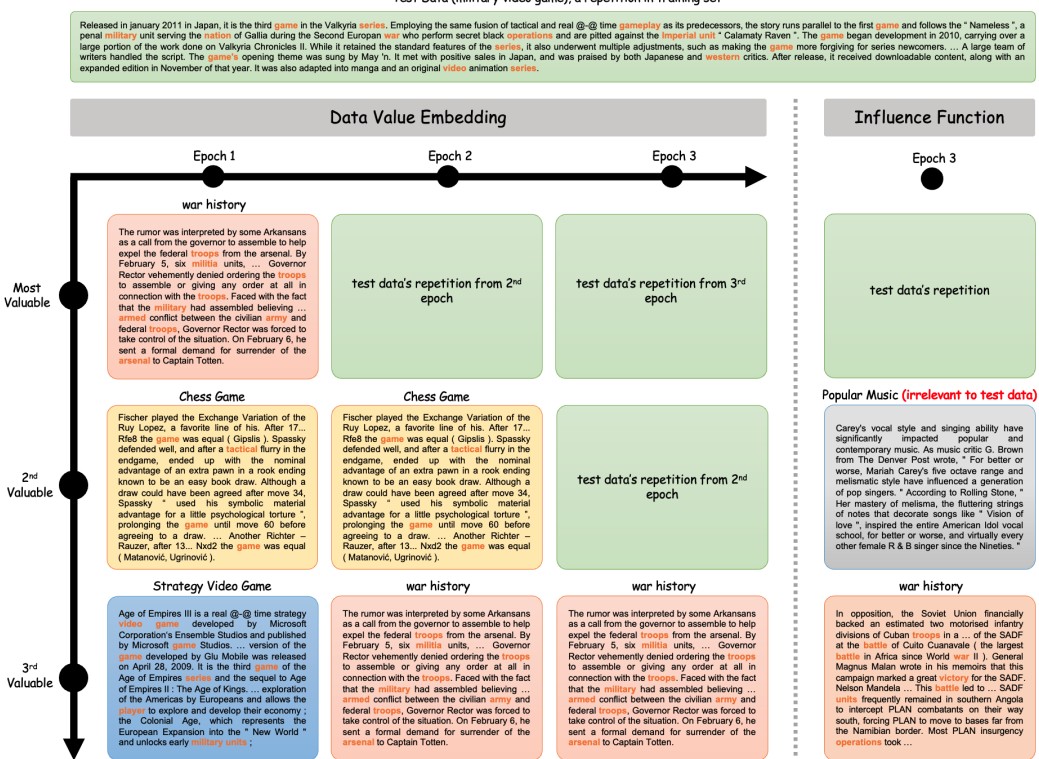

Figure 11: Visualization of (left) the evolution of the top-3 most valuable training data points identified by data value embedding throughout 3 training epochs and (right) the top-3 most valuable training data points identified by influence function. We use GPT-2 trained on Wikitext-103, with the test point being a repetition of a training data point related to a military video game. The common words between the test and training data are highlighted in  orange .

### E.5 ABLATION STUDY: ERROR FROM PROJECTION DIMENSION

We examine the error introduced by the random projection of gradient vectors, as discussed in Section 4.2. Specifically, we evaluate the Spearman correlation between data influence scores when using per-layer projection dimensions in $\{256, 1024, 2304\}$ and compare these to a larger per-layer projection dimension of $4096$. Since computing the ground truth without projection is infeasible, we use projection dimension $4096$ as a reference point for comparison. Additionally, we compare our results to LoGRA (Choe et al., 2024), the most efficient current implementation of the influence function, which also employs random projection to store data attribution information. Due to the computational and disk storage constraints, these experiments were conducted using GPT-2, trained on 5% of the Wikitext-103 dataset. The results, shown in Figure 12 (a), indicate that our data value embedding method achieves a higher Spearman correlation compared to the influence function. While our results demonstrate a clear advantage, a more in-depth analysis of the observed improvements would be an interesting direction for future research.

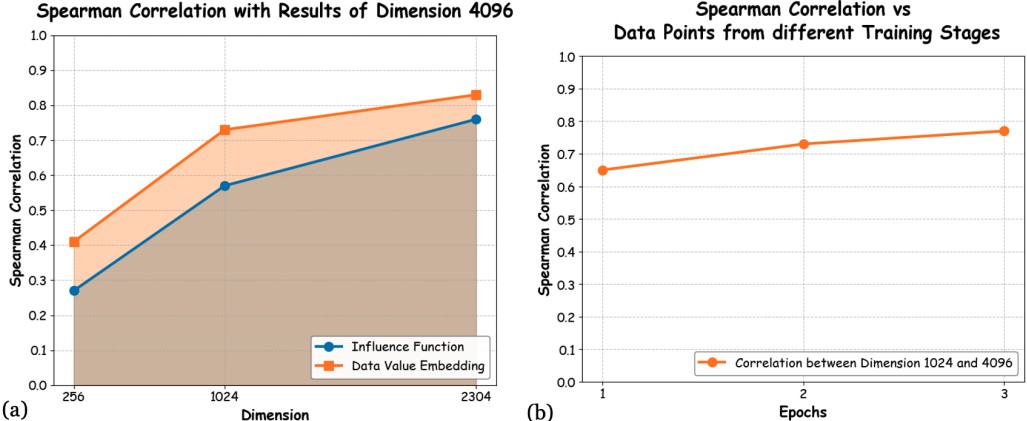

(a) (b)

Figure 12: (a) Comparison of Spearman correlation between data influence scores as a function of projection dimension.

### E.6 EXAMPLES OF BAD DATA

To demonstrate Data Value Embedding's capability in identifying potentially problematic training examples, we examined the training data points from the Pile dataset (Gao et al., 2020) that received the most negative influence scores under the same experiment settings in Section 5.3. Figure 13 and 14 show these examples and their influence scores.

Our analysis revealed several types of training data that could potentially harm model performance. First, we find quite a few code-related samples that, while syntactically valid, provide minimal educational value for language modeling. These include YAML configurations with simple numeric arrays, raw binary data represented as hexadecimal strings, and pixel-by-pixel image data in array format. Such examples contain little information about code structure or programming patterns while introducing noise into the training process.

Beyond code snippets, we found examples of text data that could potentially bias the model's learning. For instance, math problems (Figure 13's last example) that follow identical question formats could bias the model toward specific phrasings (e.g., *"What is . . ."*) rather than developing diverse language understanding. We also identified articles that, while containing meaningful content about important topics like data privacy, suffer from poor formatting with missing punctuation and paragraph breaks (Figure 14's second example). Such poorly formatted content, while topically relevant, could potentially degrade the model's ability to learn proper text formatting, punctuation usage, and document structure.

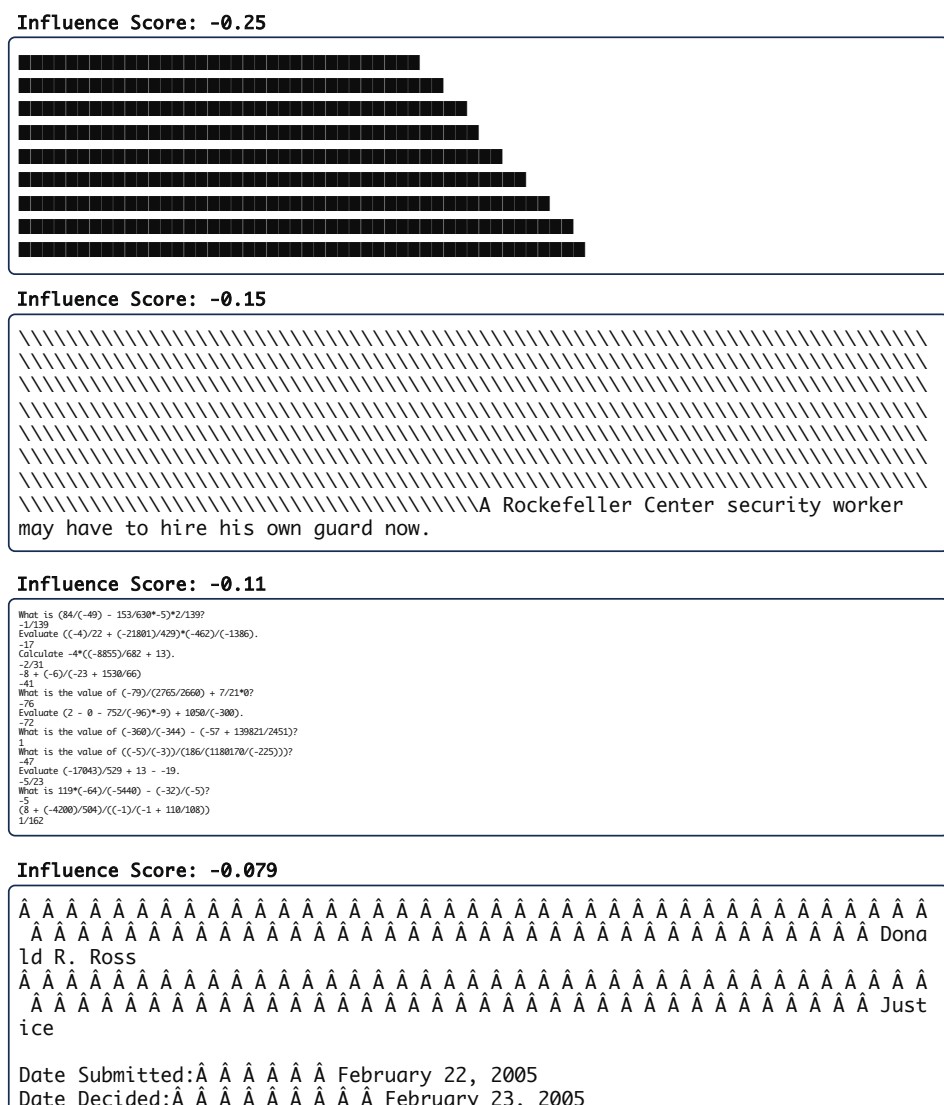

Figure 13: Examples of training data from the Pile dataset identified as potentially problematic by our method, along with their influence scores. The examples include configuration files that provide minimal learning value for language modeling, as well as repetitive mathematical problems with identical question formats.

**Influence Score: -0.078**

```
exit $status;
%YAML:1.0
x1:
  - 0
  - 4
  - 11
  - -4
  - 24
  - -3
  - -4
  - -7
  - -5
  - -2
  - 8
  - 1
  - -2
  - -5
  - -5
  - -8
  - 2
  - -8
```

**Influence Score: -0.054**

effectively they can direct them to goods and services that they are likely to buy The more companies know about their users the more competitive they are in the market Custom-tailored capitalism is what has made Google Facebook Amazon and others the richest companies in the world This profit incentive has turned big tech into a competitive field of mass intelligence gathering The better and more comprehensive the data the higher profits will be But this business model what I consider spying machines has enormous potential to violate civil liberties Big tech is already being used abroad to enhance the power of repressive regimes as my work and others has shown While it is not presently a direct threat to US democracy I worry that the potential for future abuses exists so long as big tech remains largely unregulated Big techs spy machines Current news is rife with examples of data abuses In April NBC News broke a story detailing how Facebook CEO Mark Zuckerberg had used data gathered by the platform to support his friends and defeat his rivals This is not Facebooks first privacy PR nightmare In 2018 data firm Cambridge Analytica used a Facebook app to collect data profiles of over 87 million people which was later used to distribute targeted political advertising during elections Facebook is not alone in the data collection boom This May it was revealed that Snapchat employees were using the apps data to obtain location data pictures and email addresses without users consent A new book by former Harvard business professor Shoshana Zuboff goes into great detail of the practices of what she calls surveillance capitalism Zuboff writes Once we searched Google Now Google searches us The practice goes beyond someones taste in music or what they purchase on Amazon Apps created to help people through mental illness or quit smoking sell data to big tech companies These users could be potential targets for social stigmatization or targeted advertising that exacerbates heath problems rather than solving them In December The New York Times published an expose on what one can learn about someone using their collated data from apps and smartphones By blending location tracking with other online behavior researchers were able to put together a detailed portrait of the most intimate details of users lives such as where their children go to school or who was cheating on their diet They could even tell which area of a nuclear power plant an individual worked in information that is typically classified Because of these revelations data that big tech collects poses a national security problem One open source researcher used data from Strava a fitness app to map US military bases around the world as soldiers tracked their runs Our devices are constantly telling companies where we are and what we are doing That is not always a good thing Anton Garin shutterstockcom For the worst-case scenarios look abroad Big tech is a highly unregulated sector of the economy

**Influence Score: -0.054**

```
the cancer, resulting in higher recurrence rate.1fdac852bf37e58cd7fcac827c4465e2 frame00000000
c30a05ad45c5164fbce03d62f28a7097 frame00000001
173e929536b25d8dd375f9edb5df975e frame00000002
262a928f6e1207d43bdb042011b1df21 frame00000003
2b3b51176e8e81bb0439f629c1644176 frame00000004
82ae076eba8f3a9075f8418b2f952bbe frame00000005
6b8b5b64a2c24a2496a4bde8565ebc94 frame00000006
2fd5f983139d1c5bca831915aedde5e3 frame00000007
dd4e895d4325c17a57bce5b6a9b2bcf5 frame00000008
5a0f59a2d2babc972f7442416ab008e6 frame00000009
184f8ea4bda4fb25a2a4eb56316d58fe frame00000010
06b839c9accd10708463b4fccb9658b2 frame00000011
fc6621eb8173d3dd4ae40f67c2096800 frame00000012
2c940b5b83a1e886fed3ea2402912fab frame00000013
32465d0e45fd98f790121acc2ef7adbf frame00000014
c46a77d7590c7e97c50242e87d80e6ea frame00000015
f3d886e5d11567a0523709421815ebc frame00000016
f2f30e564467a020ec37c2daf79f8f48 frame00000017
f77b9a152fffa8b69e46ae9575ee0edd frame00000018
343b484a3b963bf1a2e403b6c2854ec9 frame00000019
d7302b75ed2462bdeef092fc0dc2feeb frame00000020
```

**Influence Score: -0.047**

```
int glui_img_checkbox_1_dis[] = {     13, 13,   /* width, height */
    255,255,255,  255,255,255,  255,255,255,  255,255,255,
    255,255,255,  255,255,255,  255,255,255,  255,255,255,
    255,255,255,  255,255,255,  128,128,128,  192,192,192,
    192,192,192,  192,192,192,  192,192,192,  192,192,192,
    192,192,192,  192,192,192,  192,192,192,  192,192,192,
    255,255,255,  128,128,128,   64, 64, 64,  192,192,192,
    192,192,192,  192,192,192,  192,192,192,  192,192,192,
    192,192,192,  192,192,192,  255,255,255,  128,128,128,
     64, 64, 64,  192,192,192,  192,192,192,  192,192,192,   64, 64, 64,
    192,192,192,  192,192,192,  192,192,192,  192,192,192,
    192,192,192,  255,255,255,  128,128,128,   64, 64, 64,  192,192,192,
    192,192,192,   64, 64, 64,   64, 64, 64,   64, 64, 64,  192,192,192,
    192,192,192,  192,192,192,  192,192,192,  255,255,255,
    128,128,128,   64, 64, 64,  192,192,192,   64, 64, 64,   64, 64, 64,
     64, 64, 64,   64, 64, 64,  192,192,192,  192,192,192,
    192,192,192,  192,192,192,  255,255,255,  128,128,128,   64, 64, 64,
    192,192,192,   64, 64, 64,   64, 64, 64,  192,192,192,
     64, 64, 64,   64, 64, 64,  192,192,192,  192,192,192,  192,192,192,
    255,255,255,  128,128,128,   64, 64, 64,  192,192,192,   64, 64, 64,
```

Figure 14: Additional examples of training data identified as potentially problematic, showing text content with poor formatting (missing punctuation and paragraph breaks) and low-information code snippets with repetitive numeric arrays.

