# OpenReview forum: "Capturing the Temporal Dependence of Training Data Influence"
_ICLR.cc/2025/Conference — ICLR 2025 Oral_

### Official Review · Reviewer_tZN9 · 2024-10-26

**Soundness:** 3
**Presentation:** 3
**Contribution:** 3
**Rating:** 8
**Confidence:** 3

**Summary:**

This paper studies data influence for problems with training data not necessarily permutation invariant. Permutation invariant training data are commonly assumed in traditional data influence estimation methods. The authors proposed the concept of trajectory-specific leave-one-out (LOO) error and further proposed data value embedding to approximate trajectory LOO error. Fast computation was developed using simple dot-product between the data value embedding and the gradient of the given test data.

**Strengths:**

Many machine learning algorithms especially those for supervised learning typically assume the training data points are permutation invariant. Using such an assumption, one can divide the training data for both training and validation using strategies such as k-fold CV, leave-one-out CV etc. As the authors mentioned, many modern machine learning training paradigms especially for foundation models violate this assumption. The goal of this paper is to capture the dependence of data influence on the optimization trajectory during training. The authors proposed the trajectory-specific leave-one-out (LOO) error. Efficient computation was developed as well. Overall, the paper addresses an interesting and timely problem.

**Weaknesses:**

The problem considered in the paper is certainly interesting. However, it is important to make it clear regarding the type of CV considered in the paper. In supervised learning, CV such as LOO CV was mainly used to estimate the model performance such as prediction errors. The type considered in this paper instead is to quantify the loss change with versus without a particular data point. In this sense, it is perhaps more sensible to call it as influence of the point instead of error. The authors may reconsider the terminology to avoid confusion. If the term of LOO error remains, it is important to clarify the difference of this type and the traditional CV error.

Related to the previous point, in the traditional sense of CV error or LOO error, we would look for small CV error for model selection etc. In the sense of LOO error in the paper, the error is quantified as model’s loss change on a validation data when the training data point is removed from the training set. However, large change in the loss does not necessarily imply big differences in terms of the model performance between with versus without that particular training point. But the goal was to evaluate the model performance of that point on model trained on data without that training point. More discussion and justification are needed.
To address this issue, this review would like to suggest the authors to:

1. Explicitly compare and contrast their LOO error definition with traditional CV error
2. Discuss the implications of using loss change rather than model performance metrics
3. Provide examples or experiments showing how their LOO error relates to changes in model performance

**Questions:**

1.	How would the proposed data influence handle outliers in training data? For example, if one particular training point was an error or undesirable outlier, would the proposed influence indicate the outlier being very important since it may have big loss differences? That can be undesirable or misleading. It will be helpful if the authors could a). Conduct an experiment specifically examining the behavior of their method on datasets with injected outliers; b). Discuss potential mitigation strategies if outliers are indeed found to have outsized influence; c). Compare how their method handles outliers versus existing influence estimation techniques

2.	Using numerical examples, the authors show that early and late regions have high influence and can achieve similar performance improvements in comparison to using data from the entire process. This is quite interesting but also raises a lot of questions. Intuitively, the high influence of the early region is because one starts with nothing and the information on the early region is consequently high. After that, the added information slows down but builds up for the improvements for late region. Thus, it appears to this reviewer that one cannot simply use the early and late regions for the learning process without the middle region. Thus, more clarification and discussions on this is necessary.

3.	How would the proposed influence measure work for training time series data with seasonal and temporal dependence?

4.	Although quantification of temporal dynamics of training is interesting, the usefulness of this needs to be further explained. Can this help to detect undesirable outlier training points? Or more effective learning using a subset of training data?

---

> ### Author Response · Authors · 2024-11-24
>
> Thanks for the positive assessment!
>
> **Q [Terminology of LOO & Comparison between LOO influence and LOO CV]** *“... it is important to make it clear regarding the type of CV considered in the paper ... Explicitly compare and contrast their LOO error definition with traditional CV error”*
>
> **A** Thank you for this insightful comment about terminology. We agree that "LOO influence" more precisely describes our metric than "LOO error," and we have revised the terminology throughout the paper. We sometimes use “LOO score” in the paper.
>
> Traditional LOOCV and our LOO influence serve fundamentally different purposes. LOOCV estimates model generalization by measuring prediction performance on held-out data, serving as a model selection and evaluation technique. Lower CV errors indicate better generalization. In contrast, LOO influence quantifies how individual training points impact the model's behavior on validation data, measuring the contribution of specific training examples. Higher absolute LOO influence values indicate more impactful data points.
>
> We have added an explicit discussion in Appendix A.1 to prevent confusion. The key distinctions we will highlight include: (1) their different objectives (model evaluation vs. data importance), (2) their interpretation, and (3) their computational approaches (N separate models vs. counterfactual analysis).
>
> Thank you for helping us improve the clarity of our paper!
>
> **Q [Loss vs other performance metrics]** *“However, large change in the loss does ... (1) Discuss the implications of using loss change rather than model performance metrics (2) Provide examples or experiments showing how their LOO error relates to changes in model performance”*
>
> **A** Thank you for this thoughtful comment about the relationship between loss changes and model performance! We agree that this deserves more discussion and will address both points.
>
> **Why use validation loss instead of other metrics?**
>
> - **(1) Loss is one of the most widely used performance metrics in deep learning** and serves as the direct optimization objective during ML training, making it a natural choice in the computation of data influence quantification. This is a common choice in most of the papers in the field. In this work, we also focus on computing the LOO influence for validation loss as it is a widely accepted proxy for language model performance.
>
> - **(2) Loss values often provide richer information compared to more human-interpretable metrics like accuracy.** Consider, for example, measuring influence through classification correctness on validation data, where outcomes are binary (0 for incorrect, 1 for correct). The resulting LOO influence scores would be limited to {-1, 0, 1} for every training data point, providing far less information than the continuous values obtained from loss calculations.
>
> - **(3) The framework of Data Value Embedding supports performance metrics beyond validation loss**. The derivation in Appendix C.1 shows we can approximate the parameter difference $\theta_T - \theta_T' \approx \frac{\partial \theta_T(\varepsilon)}{\partial \varepsilon}|_{\varepsilon=0}$, allowing us to estimate $\theta_T' = \theta_T + \text{DVEmb}(z^*)$. This means that for any differentiable performance metric $f(z,\theta)$, we can approximate its LOO influence through $\nabla f(z,\theta_T)^\top\text{DVEmb}(z^*)$. While we can also evaluate non-differentiable metrics like classification accuracy through $f(z,\theta_T) - f(z,\theta_T+\text{DVEmb}(z^*))$, this approach is less efficient when computing LOO influence for all $z^*$, as it requires forward prediction on large models $N$ times where $N$ is the training data size, much slower than simply taking the dot product against each $\text{DVEmb}(z^*)$. Therefore, in this work, we focus on approximating the LOO influence for the validation loss.
>
>
> **Additional experiment:** We have conducted an additional fidelity check experiment under the same setting as Section 5.1, where we assess the correlation between LOO influence scores computed from validation loss and the ground-truth LOO score for classification accuracy. In this [figure](https://ibb.co/phpRpFJ), we observe strong positive correlations (Spearman correlation of 0.763 and 0.735) between our computed influence scores and ground-truth LOO accuracy changes in both (a) single-epoch removal and (b) all-epochs removal settings. Looking at the scatter plots, we can observe a key advantage of using loss over accuracy: while **multiple data points often share the same LOO accuracy scores** due to the discrete nature of accuracy metrics, our loss-based influence scores **provide more fine-grained distinctions between these points**. Despite this discretization effect in the ground-truth accuracy measurements, the strong correlation coefficients and clear monotonic trend in both settings demonstrate that our loss-based influence scores effectively capture changes in model performance measured by accuracy.

---

> > ### Comment · Reviewer_tZN9 · 2024-12-02
> >
> > Thanks for your clarification on the differences between the traditional LOOCV and the proposed LOO influence. This helps to avoid similar confusion for readers.

---

> > > ### Comment · Reviewer_tZN9 · 2024-12-02
> > >
> > > Given the thoughtful responses given by the authors, I improved my score from 6 to 8. I think this is a solid paper overall.

---

> ### Author Response · Authors · 2024-11-24
>
> **Q [What would be the data influence scores for outliers in the training data?]** *“How would the proposed data influence handle outliers in training data? ... It will be helpful if the authors could a). Conduct an experiment specifically examining the behavior of their method on datasets with injected outliers; b). Discuss potential mitigation strategies if outliers are indeed found to have outsized influence; c). Compare how their method handles outliers versus existing influence estimation techniques”*
>
> **A** We appreciate the reviewer's important question about handling outliers. We would like to clarify that our method naturally accounts for outliers through validation-based influence scoring. While outliers may indeed have large gradient magnitudes during training, their influence scores are determined by their impact on *validation loss*. Detrimental outliers typically result in **negative** influence scores, as they harm model performance on the validation set. This allows our method to naturally distinguish between beneficial high-influence points and harmful outliers.
>
> **In Appendix E.2.1, we directly evaluated Data Value Embedding’s ability to detect outliers through extensive experiments on mislabeled data detection.** In these experiments, we compared our approach against multiple baselines. Our method achieved strong performance in detecting mislabeled examples while demonstrating more stable results compared to retraining-based methods, as evidenced by the lower standard deviation in performance. Unlike retraining-based methods that can be unstable due to the stochastic nature of model training, our approach provides more consistent results. Moreover, by considering the entire training trajectory rather than just the final model state, our method offers more reliable outlier detection than influence functions.
>
> We have revised the main paper to make the mislabeled data detection experiment more noticeable to the readers. Thanks for the great question!
>
>
> **Q [Clarification on Figure 1(b)]** *“Using numerical examples, the authors show that early and late regions have high influence and can achieve similar performance improvements in comparison to using data from the entire process. This is quite interesting but also raises a lot of questions. Intuitively, the high influence of the early region is because one starts with nothing and the information on the early region is consequently high. After that, the added information slows down but builds up for the improvements for late region. Thus, it appears to this reviewer that one cannot simply use the early and late regions for the learning process without the middle region. Thus, more clarification and discussions on this is necessary.”*
>
> **A** We thank the reviewer for the question regarding our Figure 1. We would like to clarify an important point about Figure 1(b): we are not training the model using only data from early and late regions while skipping those from the middle region. What differs between the columns shown in Figure 1(b) is **the training phases where we apply a computationally expensive online data selection algorithm**.
>
> ​​Specifically, we adapt the online data selection algorithm from Fan et al. (2024). For each training iteration where selection is applied, we: (1) sample a candidate pool of training points of size $2B$ (where $B$ is the desired batch size), (2) compute gradient cosine similarity between each candidate point and a randomly sampled validation batch, and (3) select the $B$ points with highest similarity scores to form the next training batch. This process is computationally expensive, requiring additional computation at each selection step. Our results demonstrate that we can achieve comparable performance improvements by strategically **applying this selection process only during the early (first 2000 iterations) and late (after iteration 20000) phases of training**, while using simple random batch sampling during the middle phase.
>
> Your intuition about the importance of different training phases aligns well with our findings! The early phase is indeed critical for establishing good initialization, while the late phase is important for final model refinement. Our results suggest that **during the middle phase, when the model is in a relatively stable learning regime, the selection of high-quality training batches may not be necessary - random sampling suffices**.
>
> This insight has significant practical implications: we can allocate computationally expensive data selection efforts on the phases where they matter most, substantially reducing computational overhead without compromising model performance. We have revised the manuscript and expanded the details and discussion in Section 5.3 and Appendix E.3.2 to make our message more clear to the readers.
>
> Fan, Simin, Matteo Pagliardini, and Martin Jaggi. "DOGE: Domain Reweighting with Generalization Estimation." ICML 2024.

---

> > ### Author Response · Authors · 2024-11-24
> >
> > **Q** *“How would the proposed influence measure work for training time series data with seasonal and temporal dependence?”*
> >
> > **A** Thanks for the interesting question! Our current Data Value Embedding focuses on the regular setting. Extending it to handle time series data would require several adaptations. One possibility would be to organize training data into overlapping time blocks, similar to the approach taken by Zhang et al. (2024). Rather than treating each time point in isolation, we can consider blocks of consecutive observations. For a time series point $x_t$, we can compute its influence based on how it affects the model's predictions through each of the temporal context windows $x_t$ belongs to. We will need to develop an aggregation mechanism for the influence scores computed from each of the temporal context windows.
> >
> > We agree that the extensions to time series data are interesting future works. Thanks again for raising such an interesting discussion!
> >
> > Zhang, Yizi, et al. "TimeInf: Time Series Data Contribution via Influence Functions." arXiv preprint arXiv:2407.15247 (2024).
> >
> >
> > **Q [Usefulness of Data Value Embedding]** *“Although quantification of temporal dynamics of training is interesting ... Can this help to detect undesirable outlier training points? Or more effective learning using a subset of training data?”*
> >
> > **A** We appreciate the reviewer's question about the practical usefulness of our method. Broadly speaking, training data attribution techniques are useful for several critical aspects of AI development: (1) enhancing model interpretability and debugging capabilities, (2) enabling principled data curation and selection, and (3) providing solutions for emerging challenges in AI copyright and fair compensation. For the specific tasks of bad data detection and data selection the reviewer mentioned, we have conducted both experiments in **Appendix E.2.1**.
> >
> > **Bad data detection.** As demonstrated in our mislabeled data detection experiments (Appendix E.2.1), our method achieves strong performance in identifying bad training points compared to existing approaches. See our response to **"What would be the data influence scores for outliers in the training data?"** in the above for details.
> >
> > **Data selection.** During the rebuttal period, we have conducted additional experiments to evaluate the performance of Data Value Embedding as well as a collection of existing data attribution algorithms for the task of data selection under the same setting as our mislabeled data selection experiment. The results have been added to Appendix E.2.1. The results demonstrate that our method substantially outperforms all baseline methods across every selection budget! The superior performance of Data Value Embedding is attributed to its unique ability to capture both data quality and temporal interactions during training. *Retraining-based methods* (Data Shapley, Empirical Influence Functions, Datamodels) show limited effectiveness due to the high variance introduced by Monte Carlo sampling and learning stochasticity. While the influence function and Trak do not require model retraining, their performance is constrained by assumptions that often do not hold in practice, such as model convergence and strong convexity. KNN-Shapley provides stable valuation results. However, it assigns similar scores to similar data points, potentially reducing dataset diversity among the selected data subset. In contrast, Data Value Embedding considers both data characteristics and temporal ordering in training, allowing similar data points to receive different scores based on when they appear in the training sequence. This temporal awareness helps maintain dataset diversity while identifying valuable samples.
> >
> > Furthermore, our method's ability to track how data influence evolves throughout training provides unique insights for tasks like **curriculum learning** and **online data selection**, which can be interesting for future works.

---

> ### Author Response · Authors · 2024-11-24
> **Additional results: Qualitative Analysis of Detected Low-quality Training Data from Pile dataset**
>
> Dear Reviewer tZN9,
>
> Thank you again for your positive assessment and valuable feedback on our work. Regarding your questions about Data Value Embedding's outlier detection capabilities, we have conducted additional analyses to provide a more comprehensive response. In addition to our quantitative experiments on mislabeled data detection and data selection (Appendix E.2.1), we have also examined real-world examples from the Pile dataset that receive the most negative influence scores from Data Value Embedding (Appendix E.5). [These examples](https://plum-melicent-23.tiiny.site/) demonstrate our method's ability to detect subtle forms of problematic data. For instance, we identified code snippets containing mainly variable declarations or configuration settings that, while syntactically valid, provide minimal learning value and potentially introduce noise. We also found examples of text with poor formatting (missing punctuation and paragraph breaks) and mathematical problems with repetitive structures (frequently beginning with "What is...") that could bias the model's generation patterns.
>
> These findings demonstrate that our method can identify not just mislabeled data, but also more diverse cases that could subtly degrade model performance. This complements our quantitative evaluation in Appendix E.2.1 and provides concrete evidence of how our method handles outliers in real-world, large-scale training settings. We believe these examples provide valuable insights into the types of training data that might impede model learning.

---

### Official Review · Reviewer_Sfva · 2024-11-04

**Soundness:** 3
**Presentation:** 3
**Contribution:** 3
**Rating:** 8
**Confidence:** 2

**Summary:**

The paper proposes a novel data influence estimation method, Data Value Embedding, that captures temporal dependencies in data influence during model training. It addresses limitations of traditional influence functions by considering the order and timing of data exposure, enabling efficient real-time computation of influence scores in modern, non-convergent training settings, such as those used for large language models (LLMs).

**Strengths:**

1.	The motivation of this paper is clear: modern training paradigms—especially for foundation models using stochastic algorithms and non-convergent, multi-stage curricula—are sensitive to data ordering.
2.	It introduces a computationally efficient embedding method, making it feasible to apply influence estimation to large-scale models without retraining.
3.	Empirical results demonstrate high fidelity in data influence estimation and reveal nuanced phases in training that inform efficient data selection strategies.
4.	It provides insights on the training dynamics of foundation models: a very brief high-influence region at the start, a much longer low-influence basin, and a region in the later training stage with gradually increasing influence, resuming to a high level.

**Weaknesses:**

1.	Limited Real-World Validation: While the paper’s experiments demonstrate high fidelity on small datasets like MNIST and reduced subsets of larger datasets (e.g., 1% of the Pile), the method may not have been fully validated on more challenging, real-world datasets. This leaves questions about its robustness and scalability when applied to diverse, large-scale data used in production.
2.	Potential Overhead in Implementation: While the method reduces some computational costs, it still requires considerable storage and processing resources, particularly for storing per-sample gradient information and data value embeddings. For truly large models, such as those with billions of parameters, this may limit its practical utility without further optimizations.

**Questions:**

Could this method be applied to a closed-source model?

---

> ### Author Response · Authors · 2024-11-24
>
> We would like to thank the reviewer for the positive assessment!
>
> **Q [Robustness and scalability when applied to industrial-scale dataset]** *“While the paper’s experiments demonstrate high fidelity on small datasets like MNIST and reduced subsets of larger datasets (e.g., 1% of the Pile), ... when applied to diverse, large-scale data used in production.”*
>
> **A** We appreciate the reviewer's comment on the scalability and robustness of Data Value Embedding on large-scale datasets.
>
> **Scalability:** Our method is designed to scale efficiently to large models with minimal computational overhead. As detailed in Section 4.3 and Appendix C.9, the additional computation required is significantly lower than regular training costs ($O(BT \sqrt{p \tilde{p}}/L)$ for random projection and $O(BT \tilde{p}^2/L)$ for data value embedding computation, compared to $O(BTp)$ for standard training). Moreover, our method offers substantial efficiency improvements over existing approaches: the most efficient implementation for the influence function still requires extra compute equivalent to an additional full training run to recompute gradients on the final model. While our experimental scale was constrained by academic computing resources, our theoretical complexity analysis demonstrates that Data Value Embedding can efficiently scale to any model size that fits within available computational resources.
>
> **Robustness:** We acknowledge the challenge of extending the groundtruth comparison experiments in Section 5.1 to other datasets and models, especially for larger-scale settings. Computing ground-truth LOO scores for large models is computationally infeasible, as it requires multiple complete training runs. This is a fundamental challenge faced by all research in this field. While direct ground-truth comparison is infeasible for larger models, **we provide multiple indirect validations of our method's effectiveness at scale**. In our mislabeled data detection experiments and data selection experiments (Appendix E.2.1) using ResNet18 on CIFAR10, our method achieves competitive or superior performance compared to existing baselines. Furthermore, our qualitative analysis (Section 5.4) shows that Data Value Embedding successfully identifies semantically relevant training examples for GPT2 on Wikitext-103, while baseline methods like influence functions sometimes select irrelevant data. **While these experiments cannot definitively prove that our method perfectly approximates TSLOO in large-scale settings, they demonstrate that our approach can reliably distinguish data quality and influence for large-scale settings, which is the primary goal of data attribution.**
>
> **Q [Storage requirements]** *“Potential Overhead in Implementation: ... its practical utility without further optimizations.”*
>
> **A** We sincerely appreciate the reviewer's thoughtful comments on the storage requirements. While our method does require storage for gradient information and embeddings, we would like to emphasize two key points regarding its practical efficiency:
>
> **(1) Storage requirements are balanced by computational benefits:** Our method trades increased storage requirements for substantial computational savings. In modern computing environments, **disk storage is generally more cost-effective compared to high-performance GPU computing resources.** While storage costs are non-trivial, they represent a **one-time investment** that enables real-time data attribution capabilities - allowing immediate computation of influence scores for new test points without requiring model retraining or access to the original training data. This is particularly valuable in production environments where computational efficiency and quick response times are crucial.
>
> **(2) Dataset-level attribution aligns with practical needs while reducing storage:** In many practical applications, stakeholders are primarily interested in dataset-level rather than individual-level attribution (e.g., valuing contributions from different data providers). For these cases, as discussed in Appendix C.10, a simple extension to the derivation in Appendix C.1 shows that we can aggregate embeddings by data source, computing a single embedding that captures the collective influence of all data points from the same source. This approach not only aligns with real-world use cases but also significantly reduces storage requirements, as we only need to store one embedding per data source rather than per data point.

---

> > ### Author Response · Authors · 2024-11-24
> >
> > **Q** *“Could this method be applied to a closed-source model?”*
> >
> > **A** Thank you for this interesting question!
> >
> > Our current algorithm is primarily designed for **ML developers/trainers** who control the model training process. ML developers can integrate data value embedding into their training pipeline, and at the deployed stage, they can provide real-time attribution scores for any user queries. Such capability opens up many new possibilities, e.g., for model interpretability, model debugging, and developing fair compensation mechanisms for the current AI copyright debates where revenue can be distributed to data providers based on their measured contributions to model performance.
> >
> > If we consider the scenario where a third party wants to analyze the training data influence, our method's applicability to closed-source scenarios depends on the level of access available:
> >
> > **(1)** In scenarios with access to model checkpoints (white-box setting with access to model parameters), our method can potentially be adapted using the approach described in Section C.10 ("Approximating data value embeddings from checkpoints alone"). Specifically, we can compute gradients for each training point at each available checkpoint, assuming one gradient update step between checkpoints. While this approximation may not capture the fine-grained dynamics of the original training process, it still enables data attribution analysis when only checkpoint access is available.
> >
> > **(2)** However, in a black-box setting where only API access is available, our current method is not directly applicable as it requires gradient information. In fact, this limitation is shared by most existing data attribution methods, including influence functions, which also require access to model weights. **Currently, we are not aware of any (model-dependent) data attribution techniques that can work with only API access to the model.** We agree with the reviewer that this is an interesting question! An interesting direction for future work would be extending our approach to this setting using zero-order optimization techniques, which could potentially enable data attribution in these highly restricted scenarios.

---

> ### Author Response · Authors · 2024-11-24
> **Additional Experiments Showing Data Value Embedding's Use Case: Detecting Low-quality Training Data from Pile dataset**
>
> Dear Reviewer Sfva,
>
> Thank you again for your positive assessment and valuable feedback on our work!
>
> To follow up on your question about Data Value Embedding's use cases: while the method cannot be directly applied to closed-source models (just as other existing data attribution techniques), we believe its value for understanding training dynamics, data quality, and data influence remains significant. Our comprehensive experiments demonstrate this value through multiple analyses: studying training dynamics (Section 5.3), identifying influential training examples (Section 5.4), and detecting mislabeled data for targeted data selection (Appendix E.2.1). To further validate our method's practical usefulness, we have examined real-world examples from the Pile dataset that received the most negative influence scores from Data Value Embedding (examples are shown [here](https://plum-melicent-23.tiiny.site/) and have been added to Appendix E.5). For instance, we identified code snippets containing mainly variable declarations or configuration settings that, while syntactically valid, provide minimal learning value and potentially introduce noise. We also found examples of text with poor formatting (missing punctuation and paragraph breaks) and mathematical problems with repetitive structures (frequently beginning with "What is...") that could bias the model's generation patterns.
>
> These findings demonstrate that our method can identify not just mislabeled data, but also more diverse cases that could subtly degrade model performance. This complements our quantitative evaluation in Appendix E.2.1 and provides concrete evidence of how our method can identify bad data. We believe these examples provide valuable insights into the types of training data that might impede model learning.

---

> > ### Comment · Reviewer_Sfva · 2024-11-27
> >
> > Thank you for your detailed response. I appreciate the effort put into addressing my primary concerns. As a result of these improvements, I have decided to revise my evaluation score upward.

---

> > > ### Author Response · Authors · 2024-11-27
> > >
> > > Thank you so much for increasing the score! We are grateful for your feedback and very positive assessment of our work!

---

### Official Review · Reviewer_gYsk · 2024-11-04

**Soundness:** 3
**Presentation:** 3
**Contribution:** 3
**Rating:** 8
**Confidence:** 3

**Summary:**

This paper develops computational tools to quantify the influences of individual data points in a training session, assuming the influences are sensitive to the data order. The authors formalize the idea of trajectory-specific leave-one-out error (TSLOO), accounting for the sequence of training data. To address the computational issue of TSLOO, authors propose an efficient approximation of TSLOO using Gauss-Newton approximation. By using this technique, authors reveal distinct sages of model training such as high-impact warmup phase, low-impact basin and gradual ascending and provide an explanation.

**Strengths:**

Quantifying data influence is an important task and is crucial active sample selection. However, existing method, the influence function, does not care the order of samples' arrival, making it unsuitable for vast majority of stochastic algorithms. This paper address an important research gap.

This paper is mostly well-written. The mathematical ideas and their intuitions are clearly presented and is easy to understand.

The discovery of stages of sample influence is potentially a significant contribution to the wider machine learning community. It reveals how SGD utilize each individual sample at different stages of the training and highlights the importance of selecting data at the right time.

**Weaknesses:**

Probably due to the lack of a dedicated related work section, it is not clear where does the authors' contribution begin. For example, has TSLOO been studied before or is it a novel concept proposed by  the authors? In Section 2, the first paragraph seems to suggest that this is authors' proposal. However, a later sentence says, "while the technique of unrolled differentiation Hara et al., 2019 explicitly aims to approximate TSLOO ..." It seems the idea of TSLOO already exists in earlier works. If this is true, authors should cite existing works when introducing TSLOO at the beginning of Section 2.

Similarly, the approximation in Section 3.1 was also used by Hara et al., 2019 and is "well-established" in the literature. This makes me wonder how is the proposed work positioned among existing literature. For example, comparing to Hara et al., 2019, what is the **methodological innovation**? If everything before Data Value Embedding (DVE) is a part of literature review, the authors should make it clear.

DVEmb in Theorem 2 is an interesting idea. However, the interpretation of this quantity is unclear to me. For example,
on line 247, "this expression suggests that similar training points encountered in later iterations may have a stronger impact on the data influence score of earlier training points. " I am not sure I understand this statement.  z* are hand picked by the user so why do users care about influence score of earlier training points? Please provide a clearer explanation of this statement and why a practitioner should care about this interpretation.

on line 252, if z' is identical or highly similar to z*, their gradients will be closely aligned, leading to a significant change in DVE... I don't understand the meaning of "change" here. I guess the authors are talking about **tracking the influence of samples over the training iterations**. However, this setting isn't made clear. I suggest the authors state the context and "setting the scene" before explaining the interpretation of Theorem 2.

**Questions:**

I suggest a dedicated related work section to clearly delineate authors' novel contributions from existing concepts in the literature (e.g., Hara et al., 2019).

1. Judging from the line 127 to 134, I suppose authors consider learning of just one epoch? As in the second epoch, the counterfactual sample will again lead to the difference between theta and theta'. Do authors consider multiple epochs? and if it's just one, could authors explain why this is sufficient or how it might generalize to multiple epochs.

2. line 148, isn't Hessian H a function of the parameters?

3. line 239, it isn't clear where does the product of Hessian go at a glance. I recommend authors write briefly about the derivation of Appendix C.3.

4. Line 435, "This observation aligns with the well-established effect ..." Please cite, as it will also help a general reader (like me) to understand the background of model warm up/generalization.

5. Figure 4, " the y-axis represents the influence score of selected data points on the model at each checkpoint." How are these data points selected? It would help with reproducibility as well as the representative of the phenomenon.

also, what is the "green curve" mentioned on line 442.

---

> ### Author Response · Authors · 2024-11-23
>
> Thanks for the very positive comments about our work!
>
> **Q [Clarification on conceptual and technical contributions]** *“Probably due to the lack of a dedicated related work section, ... when introducing TSLOO at the beginning of Section 2. Similarly, the approximation in Section 3.1 was also used by Hara et al., 2019 and is "well-established" in the literature. ... If everything before Data Value Embedding (DVE) is a part of literature review, the authors should make it clear.”*
>
> **A** For *“has TSLOO been studied before or is it a novel concept proposed by the authors? …”*
>
> The basic mathematical formulation of trajectory-specific LOO was indeed first introduced as 'SGD-influence' by Hara et al. (2019). However, our work significantly extends this concept by providing the first formal treatment of trajectory-specific LOO as a fundamental framework for understanding data influence in modern deep learning. While Hara et al. (2019) focused primarily on data cleansing applications, we motivate TSLOO by highlighting why traditional permutation-invariant LOO assumptions fundamentally break down in the context of large-scale neural networks and multi-stage training curricula. Section 2 thus serves dual purposes - providing necessary background while discussing why TSLOO is more suitable for data attribution in the era of foundation models. We thank the reviewer for this observation and have revised Section 2 to better distinguish the background material from our own insights. We have cited Hara et al. (2019) when introducing TSLOO.
>
> For *“... the approximation in Section 3.1 was also used by Hara et al., 2019 and is "well-established" in the literature.”*
>
> The reviewer is correct that Section 3.1 provides preliminary (highlighted in the section title), and our core technical contribution begins in Section 3.2. While "unrolling differentiation" was first developed by Hara et al. (2019) for data cleansing, similar mathematical expressions have emerged independently in various domains, particularly in continual learning theory. However, despite its theoretical elegance, this approach has seen limited practical adoption compared to influence functions, primarily due to its substantial computational demands. For instance, Hara et al. (2019) could only analyze the final epoch of SGD training and were restricted to small model architectures. Our key contribution begins in Section 3.2, where we introduce data value embedding, a novel framework that makes the approximation from Section 3.1 computationally feasible for large-scale models. **This enables, for the first time, trajectory-specific LOO analysis at the scale of foundation models.**
>
>
> **Q [Related work section]** *“I suggest a dedicated related work section to clearly delineate authors' novel contributions from existing concepts in the literature (e.g., Hara et al., 2019).”*
>
> **A** We thank the reviewer for the great suggestion! We have an extended related work section in Appendix A, but we completely agree that incorporating a focused discussion in the main text would enhance readability and help readers better understand the context and novelty of our contributions. Following your suggestion, we have added the following paragraph to the end of Section 2.
>
> *Data attribution methods primarily fall into two categories: LOO-based methods and Shapley value-based methods. While Shapley value-based methods (Ghorbani and Zou, 2019) offer elegant theoretical interpretation, they typically require expensive model retraining, which limits their practical applicability. As a result, LOO-based methods such as influence functions (Koh and Liang, 2017) have gained more attention due to their computational efficiency. However, many studies have demonstrated that influence functions can be highly unreliable when applied to deep learning models (Basu et al., 2020; Bae et al., 2022; Epifano et al., 2023). In this work, we argue that TSLOO provides a more appropriate attribution framework for deep learning, particularly in the context of foundation models. Various research communities have independently explored Taylor expansion-based technique for approximating TSLOO for different purposes (Hara et al., 2019; Zou et al., 2021; Evron et al., 2022; Wu et al., 2022; Wu et al., 2024; Ding et al., 2024). However, practical adoption has been hindered by computational demands. In this work, we propose a new method that overcomes the computational bottlenecks in approximating TSLOO for large-scale models.*

---

> > ### Author Response · Authors · 2024-11-23
> >
> > **Q [Interpretation of Theorem 2]** *“DVEmb in Theorem 2 is an interesting idea. However, the interpretation of this quantity is unclear to me. For example, on line 247, "this expression suggests that similar training points encountered in later iterations may have a stronger impact on the data influence score of earlier training points." I am not sure I understand this statement. ... on line 252, if $z'$ is identical or highly similar to $z^{\*}$, their gradients will be closely aligned, leading to a significant change in DVE... I don't understand the meaning of "change" here. ...”*
> >
> > **A** Theorem 2 serves two key purposes.
> >
> > **(1) Computational Efficiency:** The theorem provides the theoretical foundation for our backward algorithm to efficiently compute data value embeddings. By expressing $\text{DVEmb}^{(t_s)}(z^*)$ recursively in terms of $\text{DVEmb}^{(t)}(z)$ for $t > t_s$ and $z \in B_{t}$, we can compute embeddings sequentially from the final iteration backward, significantly reducing computational costs compared to a direct implementation of Equation 2.
> >
> > **(2) Training Dynamics Insights:** Theorem 2 also provides crucial insights into how training data points interact with each other during model training. This is valuable for future research in designing training curricula or understanding training dynamics. Specifically, when two points $z^*$ and $z'$ have similar gradients, the term $\nabla \ell(\theta_t, z)^T \nabla \ell(\theta_{t_s}, z^*)$ becomes large, indicating strong interaction between these points. Consider a scenario where we're training a language model and encounter a data point $z^*$ about "quantum computing" early in training. Theorem 2 suggests that this data point's influence on the final model depends not just on the stage it is being trained on, but also on what the model sees later in training. If many similar "quantum computing" examples appear in later iterations, the influence of the original point $z^*$ on the final model may diminish! This is intuitive because later examples could have taught the model similar concepts even without $z^*$. Conversely, if $z^*$ remains one of the few examples of its kind throughout training, it maintains a higher influence.
> >
> > We appreciate the reviewer's thoughtful comments and have revised Section 4.1 accordingly to improve clarity.
> >
> > **Q** *“line 148, isn't Hessian H a function of the parameters?”*
> >
> > **A** Thanks for the catch! We have fixed the notation.
> >
> > **Q** *“line 239, it isn't clear where does the product of Hessian go at a glance. I recommend authors write briefly about the derivation of Appendix C.3.”*
> >
> > **A** Thanks! We have modified the theorem statement and relevant paragraphs accordingly.
> >
> > **Q** *“Line 435, "This observation aligns with the well-established effect ..." Please cite, as it will also help a general reader (like me) to understand the background of model warm up/generalization.”*
> >
> > **A** Thanks for the suggestion! We have added citations accordingly.
> >
> > **Q** *“Figure 4, " the y-axis represents the influence score of selected data points on the model at each checkpoint." How are these data points selected?”*
> >
> > **A** Each curve represents the average influence score across all data points from a specific iteration range. For instance, the blue curve shows the average influence scores for all data points that appeared during iterations 1000-2000 (High-impact Warmup Phase). We have updated the manuscript to make this detail more visible.
> >
> > **Q** *“what is the "green curve" mentioned on line 442.”*
> >
> > **A** Thanks for the catch! It is supposed to be “red curve” and we have fixed the manuscript.
> >
> > **Q** *“Judging from the line 127 to 134, I suppose authors consider learning of just one epoch? As in the second epoch, the counterfactual sample will again lead to the difference between theta and theta'. Do authors consider multiple epochs? and if it's just one, could authors explain why this is sufficient or how it might generalize to multiple epochs.”*
> >
> > **A** When training for multiple epochs, each appearance of a data point contributes to the model's learning trajectory. We can capture this by treating each epoch's occurrence as a distinct influence event, where removing a data point from any single epoch or all epochs creates different counterfactual training scenarios. As discussed in Appendix C.10, we can compute a data point's total influence by summing its data value embeddings across all epochs where it appears (this can be seen by a simple extension to the derivation in Appendix C.1)
> >
> > In Section 5.1, we validate data value embedding for both the scenarios of "single epoch removal" and "all-epoch removal", where the latter corresponds to exactly the scenario the reviewer mentioned (remove a data point from all epochs simultaneously). The results in Figure 3(c-d) as well as Appendix E.2 demonstrate that our method accurately approximates the ground-truth LOO in this case as well.

---

> ### Comment · Reviewer_gYsk · 2024-11-24
> **Thanks for the reply!**
>
> Thanks for the reply to my comments and for clarifing my confusions on the distinctions between this work and existing works. I will keep my score.

---

> > ### Author Response · Authors · 2024-11-24
> >
> > Thanks again for your very positive assessment and valuable feedback on our work!

---

### Official Review · Reviewer_Qbpo · 2024-11-07

**Soundness:** 3
**Presentation:** 3
**Contribution:** 3
**Rating:** 8
**Confidence:** 3

**Summary:**

This paper presents a new method called "Data value embedding" to approximate trajectory-specific Leave-One-Out (LOO) error for individual training samples, which encapsulates cumulative influence of training samples in a training trajectory. It gives some interesting insights of how influence of training batches change during a training process. The authors also provide several tips for reducing the computational cost of the method in practice.

**Strengths:**

The method introduces a novel concept by capturing data influence in a trajectory-specific manner rather than assuming permutation invariance, which is a common limitation in conventional influence estimation methods. It outlines assumptions and derives an approximation error bound, lending theoretical credibility to the approach. The approach is designed with computational efficiency, including several techniques to reduce the memory and computational cost. It enables identification of high-value data points at different stages of training, allowing practitioners to curate datasets more effectively.

**Weaknesses:**

The method is explicitly tailored for SGD and is not readily applicable to other popular optimizers like Adam. Although using SGD as a proxy is discussed, this limitation restricts the method's applicability to a broader range of models.
The evaluation with ground truth focuses on specific datasets and model types (e.g., MNIST, MLP) due to the computational cost, which may limit the generalizability of the findings.
Several assumptions are made in this paper, such as model layer independency, learning rate scheduling, which might not be satisfied and lead to reliability issue in such circumstances.

**Questions:**

1. In Line 169, the loss change is estimated by the first-order Taylor expansion, which indicates the $\theta^{'}_T$ and $\theta_T$ should be close, however, when a sample is more important, $\theta^{'}_T$ and $\theta_T$ will be more dissimilar. How this approximation can be valid in such a case?
2. Is there any trajectory pattern of the Hessian $H_k$ observed during the training process?
3. If removing the samples with negative influence scores, will a model get better performance?

---

> ### Author Response · Authors · 2024-11-23
>
> We thank the reviewer for the very positive assessment of our paper!
>
> **Q [Assumption of SGD]** *“The method is explicitly tailored for SGD ...”*
>
> **A** We appreciate the reviewer raising this important point. While our theoretical framework is derived from SGD, we would like to emphasize two key aspects:
>
> First, **the assumption about SGD is significantly less restrictive than traditional methods like influence functions**, which require strong convexity of the loss function and convergence to a global minimum. Our empirical results strongly support this: despite training models with Adam, our method successfully identifies training samples that are most relevant to the test data (Section 5.4) and achieves strong performance in mislabeled data detection and data selection benchmarks (Appendix E.2.1). These results demonstrate that our approach remains effective even when applied to models trained with SGD's variants.
>
> Second, **using SGD-based analysis as a proxy for SGD’s variants is a well-established approach in the literature**, particularly in data attribution [1,4], data selection [2,3], and data difficulty estimation [5]. This widespread adoption is because SGD's simpler update rules make it more amenable to theoretical analysis while still capturing the essential dynamics of optimization. The empirical success of this approach across various domains suggests its validity as an analytical tool.
>
> That said, we acknowledge that extending the framework of Data Value Embedding to Adam would be valuable future work. The current formulation strikes a balance between theoretical rigor and empirical feasibility, as demonstrated by our experimental results.
>
> [1] Pruthi, Garima, et al. "Estimating training data influence by tracing gradient descent." NeurIPS 2020
>
> [2] Fan, Simin et al. "DOGE: Domain Reweighting with Generalization Estimation." ICML 2024
>
> [3] Yang, Yu, et al. "Towards sustainable learning: Coresets for data-efficient deep learning." ICML 2023
>
> [4] Nguyen, Elisa, et al. "A Bayesian perspective on training data attribution." NeurIPS 2023.
>
> [5] Paul, Mansheej, et al."Deep learning on a data diet: Finding important examples early in training." NeurIPS 2021
>
>
> **Q** *“Several assumptions ... such as model layer independency, learning rate scheduling, which might not be satisfied ...”*
>
> **A** We appreciate the reviewer's attention to the assumptions in our work.
>
> **Layer Independence:** This assumption originates from the natural gradient descent literature [1] and has been widely adopted in influence function research [2, 3, 4], consistently demonstrating its effectiveness in practice. The assumption enables tractable analysis while preserving the essential characteristics of deep neural networks, as intra-layer interactions typically dominate cross-layer effects. This approximation has proven particularly valuable in developing practical algorithms for large-scale models.
>
> **Learning Rate Scheduling:** Our theoretical analysis derives error bounds for learning rate schedules of the form $O(1/\sqrt{t})$ with maximum rate $O(1/\sqrt{T})$. Though foundation model pretraining may use different schedules, these schedules share key properties with our theoretical assumptions: small magnitudes and systematic decay. Our empirical results confirm that the method's effectiveness persists even when theoretical conditions are relaxed.
>
> While these assumptions may not perfectly align with all real-world scenarios, they effectively approximate the conditions observed in our experiments. To validate this, we provide multiple empirical evaluations beyond theoretical analysis. For instance, our method demonstrates strong performance in detecting mislabeled data using ResNet18 on CIFAR10 (Appendix E.2.1), achieving comparable or better results than established baselines. The qualitative analysis (Section 5.4) shows that our method successfully identifies semantically relevant training examples, while baseline approaches sometimes select irrelevant data.
>
> We acknowledge that these assumptions could be further relaxed in future work. However, our current framework provides valuable theoretical insights while maintaining practical effectiveness, as evidenced by our comprehensive empirical results. The success of our method in real-world applications, despite potential deviations from theoretical assumptions, suggests that these approximations capture the essential aspects of data influence in deep learning systems.
>
> [1] Martens, James, and Roger Grosse. "Optimizing neural networks with kronecker-factored approximate curvature." ICML 2015
>
> [2] Grosse, Roger, et al. "Studying large language model generalization with influence functions." arXiv 2023
>
> [3] Kwon, Yongchan, et al. "DataInf: Efficiently Estimating Data Influence in LoRA-tuned LLMs and Diffusion Models." ICLR 2024
>
> [4] Choe, Sang Keun, et al. "What is Your Data Worth to GPT? LLM-Scale Data Valuation with Influence Functions." arXiv 2024

---

> > ### Author Response · Authors · 2024-11-23
> >
> > **A general remark about the comparison between Data Value Embedding and Influence Function**
> >
> > We appreciate the reviewer's thoughtful comments on the assumptions we made. In addition to our response above, we would like to additionally highlight key differences between Data Value Embedding and Influence Functions that demonstrate the relative advantages of our approach:
> >
> > **Assumptions:**
> > - While Data Value Embedding assumes SGD training with decaying learning rates (a common practice), Influence Functions require much stronger conditions of model convergence and strong convexity. These assumptions are rarely satisfied in deep learning, particularly for foundation models that often undergo just one training epoch.
> > - The layer-independence assumption and random projection techniques used in Data Value Embedding have precedent in efficient Influence Function implementations (e.g., LoGRA [1]). However, Data Value Embedding leverages these tools more effectively by integrating them into the training loop.
> >
> > **Conceptual and Computational Advantages**:
> > - Data Value Embedding explicitly captures temporal dependencies and the evolution of data influence throughout training, while Influence Functions only consider the final model state. This makes Influence Function always assigns identical influence scores to duplicate data points regardless of when they appear in training, missing crucial temporal effects.
> > - As demonstrated in Section 5.2, Data Value Embedding achieves over 15× faster computation compared to the most efficient Influence Function implementation (for the offline computation stage), while using significantly less GPU memory (0.84GB vs 63.6GB).
> > - Our experiments show that Data Value Embedding consistently outperforms Influence Function in both quantitative metrics (Section 5.1) and qualitative assessments (Section 5.4), where Influence Functions sometimes identify semantically irrelevant training examples.
> >
> > These comparisons demonstrate that while Data Value Embedding makes certain assumptions to enable scalable computation, these assumptions are either **more appropriate for deep learning applications** or **required by Influence Function's implementation as well**, while offering substantial benefits in terms of computational efficiency and capturing temporal dependencies in data influence estimation.
> >
> > [1] Choe, Sang Keun, et al. "What is Your Data Worth to GPT? LLM-Scale Data Valuation with Influence Functions." arXiv 2024

---

> ### Author Response · Authors · 2024-11-23
>
> **Q [Groundtruth comparison experiment only on specific datasets and model types?]** *“The evaluation with ground truth focuses on specific datasets and model types ... generalizability of the findings.”*
>
> **A** We acknowledge the challenge of extending the groundtruth comparison experiments in Section 5.1 to other datasets and models, especially for larger-scale settings. Computing ground-truth LOO scores for large models is computationally infeasible, as it requires multiple complete training runs. This is a fundamental challenge faced by all research in this field. **Additional experiments:** During the rebuttal time, we conducted an additional experiment using a small CNN architecture (2 convolutional layers followed by a linear layer) trained on MNIST to demonstrate our method's effectiveness beyond simple MLPs. As we can see in this [figure](https://ibb.co/hfHmptW), we observe strong correlations between our estimated influence scores and ground-truth LOO scores in both (a) single-epoch removal (Spearman correlation: 0.818) and (b) all-epochs removal (Spearman correlation: 0.682) settings. This validates our method's effectiveness across different architectures. We sincerely thank the reviewer and this additional result has been added to Appendix E.2.
>
> **Indirect validations for large-scale settings.** While direct ground-truth comparison is infeasible for larger models, we provide multiple indirect validations of our method's effectiveness at scale. In our mislabeled data detection experiments and data selection experiments (Appendix E.2.1) using ResNet18 on CIFAR10, our method achieves competitive or superior performance compared to existing baselines. Furthermore, our qualitative analysis (Section 5.4) shows that Data Value Embedding successfully identifies semantically relevant training examples for GPT2 on Wikitext-103, while baseline methods like influence functions sometimes select irrelevant data.
>
>
> **Q** “In Line 169, the loss change is estimated by the first-order Taylor expansion, ... How this approximation can be valid in such a case?”
>
> **A** We agree with the reviewer that in hypothetical scenarios where some data points are extremely influential (e.g., having exceptionally large gradients), the first-order Taylor approximation could break down as $\theta'_T$ and $\theta_T$ would become significantly different. However, in modern deep learning practice, particularly for foundation models, such scenarios are unlikely to happen as the learning rates are generally very small. In addition, gradient clipping is also a commonly used technique in foundation model pretraining. These methods effectively prevent any single data point from causing dramatic parameter changes.
>
> Furthermore, our empirical validation supports the effectiveness of this approximation in practical ML training scenarios: (1) In addition to the high correlation with ground-truth TSLOO scores in Section 5.1 and Appendix E.2, Data Value Embedding achieves strong performance on the standard benchmarks of mislabeled data detection and data selection in Appendix E.2.1. (2) In the qualitative evaluation in Section 5.4, our method successfully identifies training data that are semantically correlated with the test data point.
>
> Overall, while we acknowledge the theoretical possibility raised by the reviewer, in training setups typical of modern deep learning, our approximation provides a practical and reliable way to assess data influence.

---

> > ### Author Response · Authors · 2024-11-23
> >
> > **Q** “Is there any trajectory pattern of the Hessian $H_k$ observed during the training process?”
> >
> > **A** ​​Computing and storing the full Hessian matrix is computationally intractable for modern neural networks, as it requires $O(p^2)$ memory and computation where $p$ is the number of parameters. Therefore, following standard practice in optimization literature [1, 2], we approximate the Hessian using the Generalized Gauss-Newton (GGN) matrix: $H_k \approx G_k = \sum_{z\in B_k} \nabla\ell(\theta_k,z)\nabla\ell(\theta_k,z)^\top$. The effectiveness of this approximation is validated through our fidelity experiments in Section 5.1, where we demonstrate strong correlation with ground-truth LOO scores despite using the GGN approximation.
> >
> > While tracking the full spectrum of $G_k$ remains challenging, we can gain insights by analyzing gradient norm trajectories. Under the assumption of approximately orthogonal gradients within batches (which is reasonable given the high dimensionality of the parameter space), the eigenvalues of $G_k$ are primarily determined by squared per-sample gradient norms. As shown in this [figure](https://ibb.co/TkS0y1W) (obtained under the same setting as in Section 5.3), we observe that gradient norms start large during early training and gradually stabilize, leading to corresponding changes in the eigenvalues of $G_k$. This evolution directly shapes the influence dynamics through the term $(I - \eta_tG_t)$ in our data value embedding formulation, as we discussed in Section 5.3 and Appendix E.3.1.
> >
> > Thank you for raising this interesting question about Hessian trajectories! While our current work focuses on developing an efficient method for capturing temporal dependencies in data influence, a detailed analysis of Hessian dynamics during training is an important research direction for sure. There are interesting recent works specifically focused on this topic [3, 4], which provide valuable insights into the evolution of loss geometry during neural network training.
> >
> > [1] Martens, James, and Roger Grosse. "Optimizing neural networks with kronecker-factored approximate curvature." International conference on machine learning. PMLR, 2015.
> >
> > [2] Martens, James. "New insights and perspectives on the natural gradient method." Journal of Machine Learning Research 21.146 (2020): 1-76.
> >
> > [3] Wang, Zixuan, Zhouzi Li, and Jian Li. "Analyzing sharpness along GD trajectory: Progressive sharpening and edge of stability." Advances in Neural Information Processing Systems 35 (2022): 9983-9994.
> >
> > [4] Song, Minhak, and Chulhee Yun. "Trajectory Alignment: Understanding the Edge of Stability Phenomenon via Bifurcation Theory." Advances in Neural Information Processing Systems 36 (2024).
> >
> > **Q** *“If removing the samples with negative influence scores, will a model get better performance?”*
> >
> > **A** Thanks for the insightful question! We have conducted additional experiments to evaluate the performance of Data Value Embedding as well as a collection of existing data attribution algorithms for the task of data selection under the same setting as our mislabeled data selection experiment. The results have been added to Appendix E.2.1. The experimental results demonstrate that our method substantially outperforms all baseline methods across every selection budget! The superior performance of Data Value Embedding is attributed to its unique ability to capture both data quality and temporal interactions during training. *Retraining-based methods* (Data Shapley, Empirical Influence Functions, Datamodels) show limited effectiveness due to the high variance introduced by Monte Carlo sampling and learning stochasticity. While the influence function and Trak do not require model retraining, their performance is constrained by assumptions that often do not hold in practice, such as model convergence and strong convexity. KNN-Shapley provides stable valuation results. However, it assigns similar scores to similar data points, potentially reducing dataset diversity among the selected data subset. In contrast, Data Value Embedding considers both data characteristics and temporal ordering in training, allowing similar data points to receive different scores based on when they appear in the training sequence. This temporal awareness helps maintain dataset diversity while identifying valuable samples.

---

> > > ### Comment · Reviewer_Qbpo · 2024-11-26
> > > **Thanks for authors' response**
> > >
> > > Thanks for authors' comprehensive response. I think the additional experimental results are quite interesting and demonstrate effectiveness of the proposed method. I will keep my score for recommending acceptance of the paper.

---

> > > > ### Author Response · Authors · 2024-11-26
> > > >
> > > > We sincerely thank you for taking the time to review our rebuttal. We are very grateful for the endorsement of our work!

---

### Meta-Review · Area_Chair_ZxnH · 2024-12-17

**Metareview:**

Quantifying data influence is an important task. However, existing methods are permutation invariant, making them unsuitable for the vast majority of stochastic algorithms for nonconvex optimization. This paper addresses an important research gap.

The paper is mostly well-written. The mathematical ideas and their intuitions are clearly presented.

The discovery of stages of sample influence is potentially a significant contribution to the wider machine learning community. It reveals how SGD utilizes each individual sample at different stages of the training and highlights the importance of selecting data at the right time.

There is consensus among reviewers that this paper should be accepted for publication.

Minor remark: I was surprised that in the SGD equations, the learning rate was not normalized by the batch size (as done for instance in Hara et al). I suppose this normalization can be incorporated into the learning rate but it would be good to add a remark about this.

**Additional Comments On Reviewer Discussion:**

Authors addressed all reviewer comments.

---

### Decision · Program_Chairs · 2025-01-22

Accept (Oral)